# IRF2 is a master regulator of human keratinocyte stem cell fate

Nicolas Mercado [1], Gabi Schutzius [1], Christian Kolter [1], David Estoppey [1], Sebastian Bergling [1], Guglielmo Roma [1], Caroline Gubser Keller[1], Florian Nigsch [1], Adrian Salathe [1], Remi Terranova [2], John Reece-Hoyes[3], John Alford[3], Carsten Russ[3], Judith Knehr[1], Dominic Hoepfner [1], Alexandra Aebi [1], Heinz Ruffner[1], Tanner C. Beck[4], Sajjeev Jagannathan[4], Calla M. Olson[4], Hadley E. Sheppard[5], Selma Z. Elsarrag[5], Tewis Bouwmeester[1], Mathias Frederiksen [1], Felix Lohmann[6,7]*, Charles Y. Lin [4,5,7]* & Susan Kirkland [1,7]*

Resident adult epithelial stem cells maintain tissue homeostasis by balancing self-renewal and differentiation. The stem cell potential of human epidermal keratinocytes is retained in vitro but lost over time suggesting extrinsic and intrinsic regulation. Transcription factor-controlled regulatory circuitries govern cell identity, are sufficient to induce pluripotency and transdifferentiate cells. We investigate whether transcriptional circuitry also governs phenotypic changes within a given cell type by comparing human primary keratinocytes with intrinsically high versus low stem cell potential. Using integrated chromatin and transcriptional profiling, we implicate IRF2 as antagonistic to stemness and show that it binds and regulates active *cis*-regulatory elements at interferon response and antigen presentation genes. CRISPR-KD of IRF2 in keratinocytes with low stem cell potential increases self-renewal, migration and epidermis formation. These data demonstrate that transcription factor regulatory circuitries, in addition to maintaining cell identity, control plasticity within cell types and offer potential for therapeutic modulation of cell function.

[1] Chemical Biology & Therapeutics, Novartis Institutes for Biomedical Research, Novartis Pharma AG, Forum 1 Novartis Campus, CH-4056 Basel, Switzerland. [2] Preclinical Safety, Translational Medicine, NIBR, Basel CH-4057, Switzerland. [3] Chemical Biology & Therapeutics, Novartis Institutes for Biomedical Research, 250 Massachusetts Avenue, Cambridge, MA 02139, USA. [4] Therapeutic Innovation Center, Department of Biochemistry and Molecular Biology, Baylor College of Medicine, 1 Baylor Plaza, Houston, TX 77030, USA. [5] Department of Molecular and Human Genetics, Baylor College of Medicine, 1 Baylor Plaza, Houston, TX 77030, USA. [6] Autoimmunity, Transplantation and Inflammation, NIBR, Basel CH-4056, Switzerland. [7] These authors jointly supervised: Charles Y. Lin, Felix Lohmann and Susan Kirkland. *email: Felix.Lohmann@novartis.com; Charles.Y.Lin@bcm.edu; Susan.Kirkland@novartis.com

Tissue stem cells (SCs) maintain homeostasis and repair damage throughout life. Stem cells reside in specialized niches[1] which preserve and balance stem cell renewal with delivery of differentiated cells. The best understood adult tissue stem cell, the hematopoietic stem cell (HSC), resides in the bone narrow niche and supplies all blood cell lineages[2]. HSCs are distinct cell types which can be isolated and shown to renew the entire blood system experimentally and clinically with transplantation therapies. The HSC paradigm drove the search for similarly rare "hard-wired" epithelial stem cells[3]. However, recently the existence of equivalent slow cycling quiescent epithelial stem cells has been questioned[4], leading to the proposal that shifting focus from stem cell phenotype to function would better enable advances in epithelial stem cell biology[5].

The Epidermis, the outermost layer of skin comprised mainly of keratinocytes, acts as a physical barrier and is replaced every few weeks by resident stem cells residing in the basal layer[6,7]. Cultivation of human keratinocytes in vitro was first achieved using 3T3 feeders[8] and is now enabled with specialist serum-free culture media. Epidermal stem cell function is retained in vitro, albeit with a high degree of heterogeneity suggesting some intrinsic regulation, and used therapeutically in burns patients. Furthermore, in vitro expanded transgenic epidermal stem cells were recently used to replace an entire epidermis in life-threatening epidermolysis bullosa, supporting the existence of a long-lived population of self-renewing epidermal stem cells[9].

Recent evidence demonstrates that the integration of niche signals with transcriptional circuitries specifies stem cell fate decisions[10], arguing for extrinsic and intrinsic maintenance of stemness[11]. A role for epigenetic factors and individual TFs has been demonstrated in epithelial stem cell fate decisions[11] and transcriptional enhancer clusters that specify cell identity, termed super-enhancers[12], have been shown to be involved in lineage decisions in hair follicle stem cells[13]. As transcription factors (TFs) have a pivotal role in mediating cell state, exemplified in the discovery of induced pluripotential stem cells[14], we set out to identify the transcriptional landscape specifying the epidermal stem cell state.

TFs recruit co-activators like BRD4, mediator, and histone acetyltransferases to genes in order to drive their expression. We and others have shown that in any given cell type, cell identity is specified by a highly interconnected regulatory network of TFs termed core regulatory circuitry (CRC)[15–17]. CRC TFs often physically interact at the protein–protein level[18,19], are regulated by large enhancers or super-enhancers[20], and tend to form highly interconnected regulatory circuits by binding to the promoters and enhancers of other CRC TFs[21]. Although such transcriptional circuitries define cell identity, they have not been used to differentiate cell states within a single cell type.

Here, we define the transcriptional network maintaining the human epidermal stem cell state using comprehensive transcriptional and epigenomic profiling, computational circuitry analysis and genetic validation. To generate cell numbers necessary for such an unbiased analysis, we employed replication-induced loss of stem cell function as our model system, supported by previous studies showing concomitant replicative senescence and differentiation in this in vitro model and in vivo in aged mouse gingival epithelium[22]. Multiple phenotypic read-outs confirm high and low stem cell potential in our populations. Epidermal stem cells are more clonogenic[8], migratory[23], adherent to extracellular matrix (ECM)[24] and better able to form a stratified epidermis in 3D skin models. We delineate chromatin-mediated cis-regulation associated with loss of stem cell function and identify putative TFs acting as master regulators of keratinocyte stem cell function, several of which are validated using CRISPR-mediated genetic loss of function. Most

significantly, we show that loss of IRF2 induces a rapid, dramatic, and sustained enhancement of stem cell function in human keratinocytes.

CRC analysis identifies transcriptional master regulators in cell state transitions supporting the concept of a "hard-wired" stem cell circuitry where interconnectivity of TF network maintains stem cell function. Clearly, environmental factors will be expected to intersect with this circuitry to define the final ability of stem cells to retain their self-renewal state. Such understanding opens up possibilities for therapeutic intervention in epithelial regeneration.

## Results

**Keratinocytes steadily lose stem cell potential in vitro.** Primary human keratinocytes lose stem cell function over time in vitro, and were characterized at early passage 7 as High Stem Cell Potential-Human Keratinocytes (HSCP-HKs) and after further passaging until passage 15 as Low Stem Cell Potential-Human Keratinocytes ("LSCP-HKs") (Fig. 1a). HSCP-HKs generated significantly more and larger colonies (Fig. 1b), migrated faster (Fig. 1c), and attached either more quickly (Collagen I) or in larger numbers (Collagen IV, Fibronectin and Laminin V) than LSCP-HKs (Fig. 1d), supporting higher stem-cell function. Serial passaging of primary cells induces senescence[25], which we reproduced in our system as confirmed by an increase in Senescence-Associated Beta-Galactosidase (SA-β-Gal) staining in LSCP-HK cells (Fig. 1e). Transcriptomic analysis (RNA-seq) and comparison of LSCP-HKs with HSCP-HKs shows enhanced expression of 178 genes and decreased expression of 416 genes in LSCP-HKs (Fig. 1f, Supplementary Data 1). Gene set analysis (GSA) of the HSCP-HK transcriptome shows enrichment of genes categorized as "Cell Cycle/Mitotic", "DNA Replication", and "Cell Adhesion Molecules" (Fig. 1g, Supplementary Fig. 1a, Supplementary Data 2), exemplified by known cell cycle regulators (CCNA2, AURKB, E2F1, and UBE2C) and supporting their higher proliferation and ECM-binding affinity (Fig. 1g, Supplementary Fig. 1a, Supplementary Data 2). LSCP-HKs were enriched for categories related to "Immune Response" and "Inflammatory Response" with upregulation of CXCL8, TNF, NFKBIZ, PTGS2, S100A9, and IL1B (Fig. 1g, Supplementary Fig. 1b, Supplementary Data 2), a transcriptome with similarities to other senescent phenotypes in line with the increased SA-β-Gal staining[26]. Reduced cell proliferation in LSCP-HKs accompanied decrease in cell cycle promoting genes, CCNA2 and UBE2C, and concomitant increase in negative regulators such as CDKN2A (p16INK4a) but not CDKN1A (p21WAF1), as previously reported in passaging induced senescence[27]. Generation of reactive oxygen species and ECM degradation was inferred by upregulation of anti-oxidant response genes such as NQO1, SOD2, GPX2, and PTGS2 and by matrix metalloproteinases (MMP1,-3,-9) all of which are again associated with stress-induced premature senescence[26] (Supplementary Fig. 1c). In addition, an increase in pro-inflammatory chemokines and cytokines including CXCL8, TNF, IL1A, and IL1B suggests a possible senescent associated secretory phenotype[28]. Finally, FOXM1, a member of the Forkhead superfamily of TFs, previously associated with high proliferative potential in keratinocytes, was decreased in LSCP-HKs[29] (Supplementary Fig. 1c).

Metacore TF connectivity analysis[30] showed an association of LSCP-HKs with pro-inflammatory (NF-kB family) and Interferon signaling TFs (STAT1 and IRF1), and HSCP-HKs with pluripotency-associated TFs (NANOG, Oct-3/4, and SOX2), cell cycle regulators (E2F1 and E2F4) and p63, a TF regulator of epithelial stem cell compartments[31] (Fig. 1h).

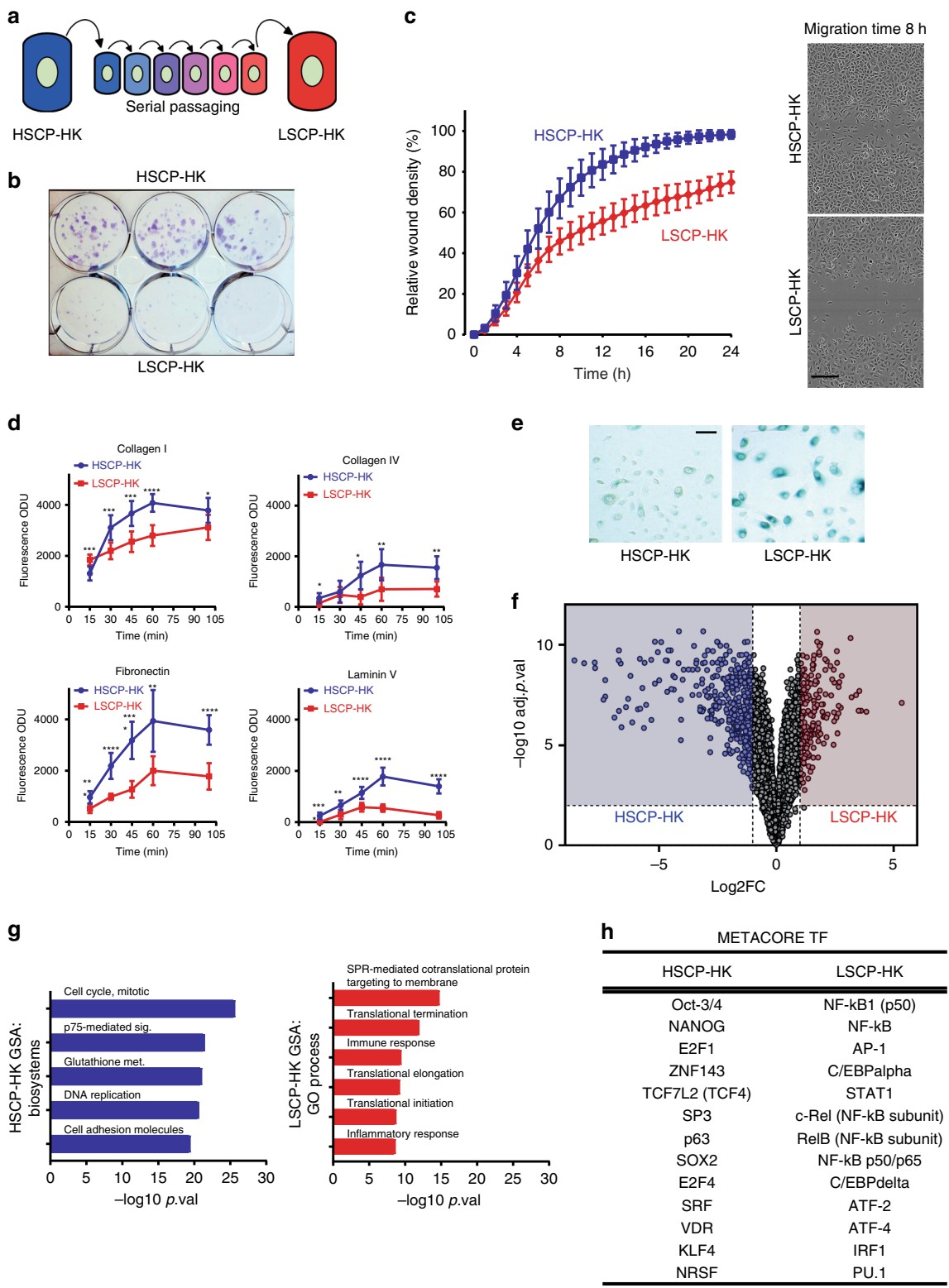

**BRD4 is dynamically distributed between HSCP-HK and LSCP-HK.** Differential gene expression and stem cell function between HSCP-HKs and LSCP-HKs suggests a change in cell state which is imposed by a cohort of specific TFs and epigenetically maintained[15,32]. We contrasted chromatin landscapes between HSCP-HKs and LSCP-HKs by genome-wide profiling of H3K27ac, a histone mark found at active promoters and enhancers[12,33–35], and of BRD4, a chromatin co-activator directly

associated with active transcription elongation[36], as well as of chromatin accessibility to identify nucleosome free *cis*-regulatory elements likely to harbor TF binding sites[37].

Using H3K27ac, we defined 24,614 discrete regions of active chromatin present in at least one sample in HSCP-HKs or LSCP-HKs. Although unbiased clustering of H3K27ac occupancy profiles segregated HSCP-HK and LSCP-HK replicates ($n = 3$), overall the active chromatin landscapes were highly similar

**Fig. 1** Serial passaging of keratinocyte stem cells induces loss of "stemness". **a** Schematic depicting "HSCP-HKs" transformation into "LSCP-HK" by serial passaging. **b** Colony formation assay comparing HSCP-HKs to LSCP-HK after 8 days of culture (2000 cells/well). **c** Migration assay comparing HSCP-HKs to LSCP-HKs after "wounding" a confluent monolayer of cells. Representative images of HSCP-HKs and LSCP-HKs at 8 h time point are shown. Scale bar = 300 μm. **d** Adhesion assay comparing HSCP-HKs to LSCP-HKs with regard to attachment to Collagen I, Collagen IV, Fibronectin, or Laminin V-coated plastic in vitro. Time (min) plotted versus Fluorescence optical density units (ODU) indicative of cell number. Means ± S.D. of $n = 7$ technical replicates for Collagen IV and $n = 8$ technical replicates for Collagen I, Fibronectin, and Laminin V. One-way ANOVA with Holm-Šídák multiple comparisons test. $^*p < 0.05$, $^{**}p < 0.01$, $^{***}p < 0.001$, $^{****}p < 0.0001$. **e** Beta-galactosidase staining as a marker of cellular senescence comparing HSCP-HKs to LSCP-HKs in culture. Scale bar = 200 μm. **f** Global difference in gene expression (RNA-seq) between HSCP-HKs and LSCP-HKs, plotting fold change (FC) versus adjusted $p$ value (adj.p.val). Differentially expressed genes are marked in red for LSCP-HK (178 genes total) versus in blue for HSCP-HK (416 genes total) with significance cut-offs defined as absolute value log2FC > 1 and −log10 adj.pval > 2. **g** Gene sets most enriched in HSCP-HKs (Biosystems) or LSCP-HKs (GO Process) ranked by $p$ value (p.val). **h** List of transcription factors (TFs) from MetaCore Interactome enrichment tool comparing HSCP-HKs and LSCP-HKs based on global gene expression data

(Supplementary Fig. 2a) and likely reflect keratinocyte identity. This suggests that stem cell potential arises from subtle alterations of chromatin landscape as opposed to de novo formation of euchromatin or heterochromatin.

Previously, we demonstrated that TFs can redistribute the transcriptional co-activator BRD4 across chromatin landscapes to reshape both chromatin and gene expression[38]. Across active genes ($n = 7278$) in both HSCP-HK and LSCP-HK, we quantified BRD4 total occupancy (area under the curve, AUC) at all promoter and proximal enhancer elements (<50 kb from the transcription start site (TSS)). Ranking active genes by the change in BRD4 occupancy between HSCP-HK and LSCP-HK (Fig. 2a), revealed a strong association with changes in chromatin acetylation and gene expression both globally (Fig. 2a) and at individual exemplary genes (Fig. 2b, c). Notably, at genes where BRD4 was redistributed, H3K27ac chromatin was present in both HSCP-HKs and LSCP-HKs (Fig. 2c) suggesting that despite similarities in active chromatin landscapes, redistribution of transcriptional co-activators like BRD4 may play an important role in modulating stem cell potential.

**Circuitry analysis suggests potential TF master regulators**. To quantify the role of individual TFs in reshaping the keratinocyte active chromatin landscape, we first prioritized a cohort of TFs possessing features of CRC TFs (Fig. 3a). Actively expressed TFs were filtered for the presence of a known binding motif[39–41], regulation by an enhancer element, and evidence of protein–protein interaction with other TFs[42]. This resulted in 60 remaining TFs that via unbiased Markov Chain Linkage clustering segregated into six distinct clusters with two predominating clusters. Compared to other actively expressed TFs, these 60 displayed higher regulatory connectivity (binding to each other's regulatory regions) (Supplementary Fig. 3a–c). Interestingly, these two large clusters contained TFs with higher inter-cluster regulatory connectivity (Supplementary Fig. 3d) and with biological functions consistent with thematic HSCP-HK or LSCP-HK gene expression and Metacore signatures (e.g., growth/proliferation for HSCP-HK and inflammatory signaling for LSCP-HK) (Fig. 3a). Based on these properties of enhancer regulation, high regulatory, and protein–protein interactions, and HSCP-HK and LSCP-HK associated functions, we hypothesized that these two clusters represent the CRC TFs that underlie stem cell potential in keratinocytes.

Using motif enrichment finding with FIMO[43], CRC TFs were assigned to putative nucleosome free regions (as defined by ATAC-seq). Clustering of CRC TF motif occupancy revealed modules with similar or contrasting binding patterns (Supplementary Fig. 3e). Interestingly, IRF TFs showed a highly dissimilar pattern of occupancy to a set of basic helix loop helix (bHLH) TFs (MYC, MAX, SREBF1/2, and BHLHE40) and a set of promoter associated housekeeping factors (E2F4, MAZ, YY1,

SP1, TFAP2A, KLF5, and KLF13), both of which associated with the HSCP-HK cluster 1. These data suggested that the HSCP-HK and LSCP-HK associated clusters of TFs have distinct targets and gene-expression programs.

To measure the effect of individual CRC TFs on the keratinocyte chromatin landscape, we quantified changes in BRD4 occupancy proximal to predicted nucleosome free binding sites as a measure of TF outward effect (BRD4 OUT Degree) (Fig. 3b, c). We next ranked all 60 CRC TFs by change in BRD4 OUT degree between HSCP-HK and LSCP-HK. This ranking revealed a gradient of TF activity delineating BRD4 changes between HSCP-HKs and LSCP-HKs TFs. TFs regulating growth and proliferation such as MYC, MAZ, and E2F4 drove increased BRD4 in HSCP-HKs and conversely, pro-inflammatory TFs including the IRF family drove increased BRD4 in LSCP-HKs. Based on the ability of these CRC TFs to drive BRD4 re-localization between HSCP-HKs and LSCP-HKs, we hypothesized that these TFs are not only associated with HSCP-HK versus LSCP-HK identity, but are functionally required for maintenance of keratinocyte stem cell function (Fig. 3d).

**CRISPR–Cas9 screen identifies function modulating TFs**. To validate the role of CRC predicted TFs in keratinocyte stem cell function, we performed a pooled CRISPR–Cas9 screen and sampled the population over time. Keratinocytes with greater stem cell properties proliferate faster and survive longer in vitro[44], therefore single-guide RNA (sgRNAs) targeting genes promoting stem cell function should be quickly lost from the pool while sgRNAs to genes antagonistic to stem cell function should persist. The screen was performed in neonatal keratinocytes which, unlike adult keratinocytes, were sufficiently robust for infection and selection of a Cas9 expressing population essential for such a pooled screen (Supplementary Fig. 4). The pool contained 2698 sgRNAs targeting 540 genes, including 34 out of the 60 putative TF regulators from the CRC analysis (Supplementary Fig. 4b), along with predicted stem cell regulators and controls. After infection at day 0 and selection with puromycin for 3 days, cell pellets were collected for DNA sequencing at days 12, 19, 28, 33, 38, 43, and 47 post infection (Fig. 4a). SgRNA barcode counts over time showed a rapid decrease with most barcodes lost at day 38 (Supplementary Fig. 5a, b) suggesting that Cas9 expressing keratinocyte proliferation was limited regardless of the effects of editing. Evidence from positive control sgRNAs targeting essential genes supported this suggestion showing only modest effects at days 12 and 19 (Supplementary Fig. 5c). Therefore, negative effects of editing on proliferation were evaluated early (days 12 and 19) while increased survival and cell proliferation were assessed later (days 38 and 43) (Fig. 4b, Supplementary Data 3). When evaluating the top seven most strongly represented CRC-associated TFs by significance (redundant siRNA activity (RSA) Down/Up) and magnitude (Q1/3), we observed an early

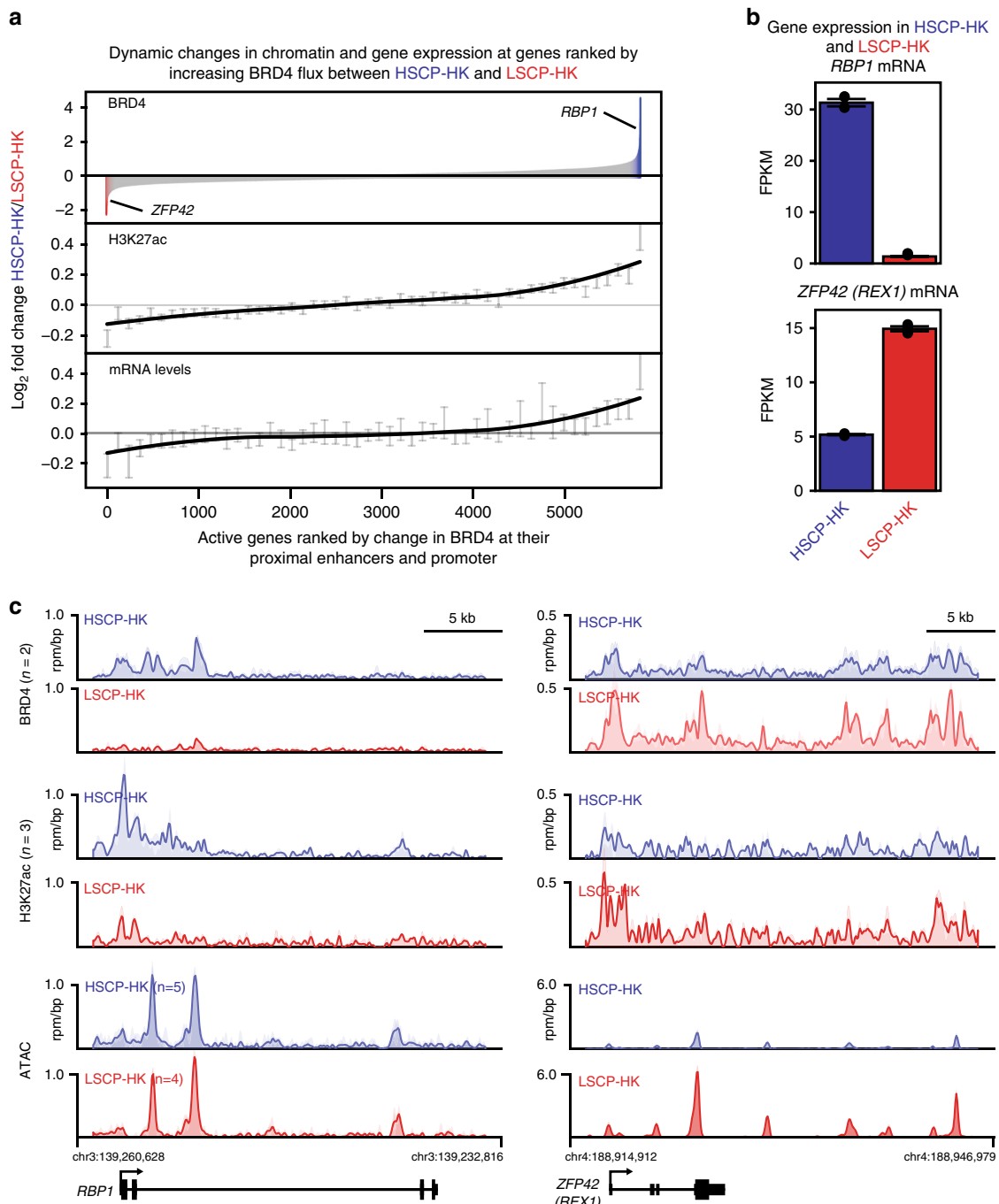

**Fig. 2** Dynamic redistribution of BRD4 between HSCP-HKs and LSCP-HKs. **a** Ranking of all active genes by change in BRD4 occupancy at their promoter and proximal enhancers between HSCP-HK and LCSP-HK, with fold change in BRD4 occupancy shown in top panel. In middle and bottom panels, fold change in H3K27ac and mRNA levels are shown using a binned average (100 genes per bin), respectively. Error bars represent 95% confidence intervals (CI) of the average as determined by resampling with replacement (1000 permutations). A trend line (black) is overlaid by lowest locally weighted regression. **b** Transcript levels of RBP1 and ZFP42 based on RNA-seq (FPKM) comparing HSCP-HK to LSCP-HK. Error bars represent standard deviation. **c** Chromatin occupancy of BRD4 and H3K27ac (ChIP-seq) as well as chromatin accessibility (ATAC-seq) at RBP1 and ZFP42 genomic loci as read density in units of reads per million per base pair (rpm/bp). Individual HSCP-HK (blue) and LSCP-HK (red) replicates are shown as shaded traces. The average of replicates is drawn as a solid line

"drop-out" of HSCP-HK TFs (MYC, E2F4, MAZ, TEAD1, and NFE2L2), indicative of a role in maintaining stem cell function (Fig. 4b). E2F4 and MYC, within the top three CRC candidates, are known positive regulators of keratinocyte proliferation[45,46] (Fig. 3d). MYC was amongst the top TFs whose editing drastically reduced cell proliferation (from day 12 to day 28) whereas a "neutral" CRC TF, SNAI2, was shown to be among the top TFs

from day 19 onwards (Supplementary Fig. 5d, e). SNAI2 is known to repress expression of differentiation genes[47]. At days 38 and 43, we also found an overrepresentation of sgRNAs to LSCP-HK TFs (IRF6, RELA, STAT1, ERF, TGIF1, VDR, RUNX1, ESSRA, and IRF2) implicating these CRC-TFs in the loss of stem cell phenotype with time in vitro (Fig. 4b). Of particular interest was the strong enrichment of sgRNAs targeting IRF2 from day 28

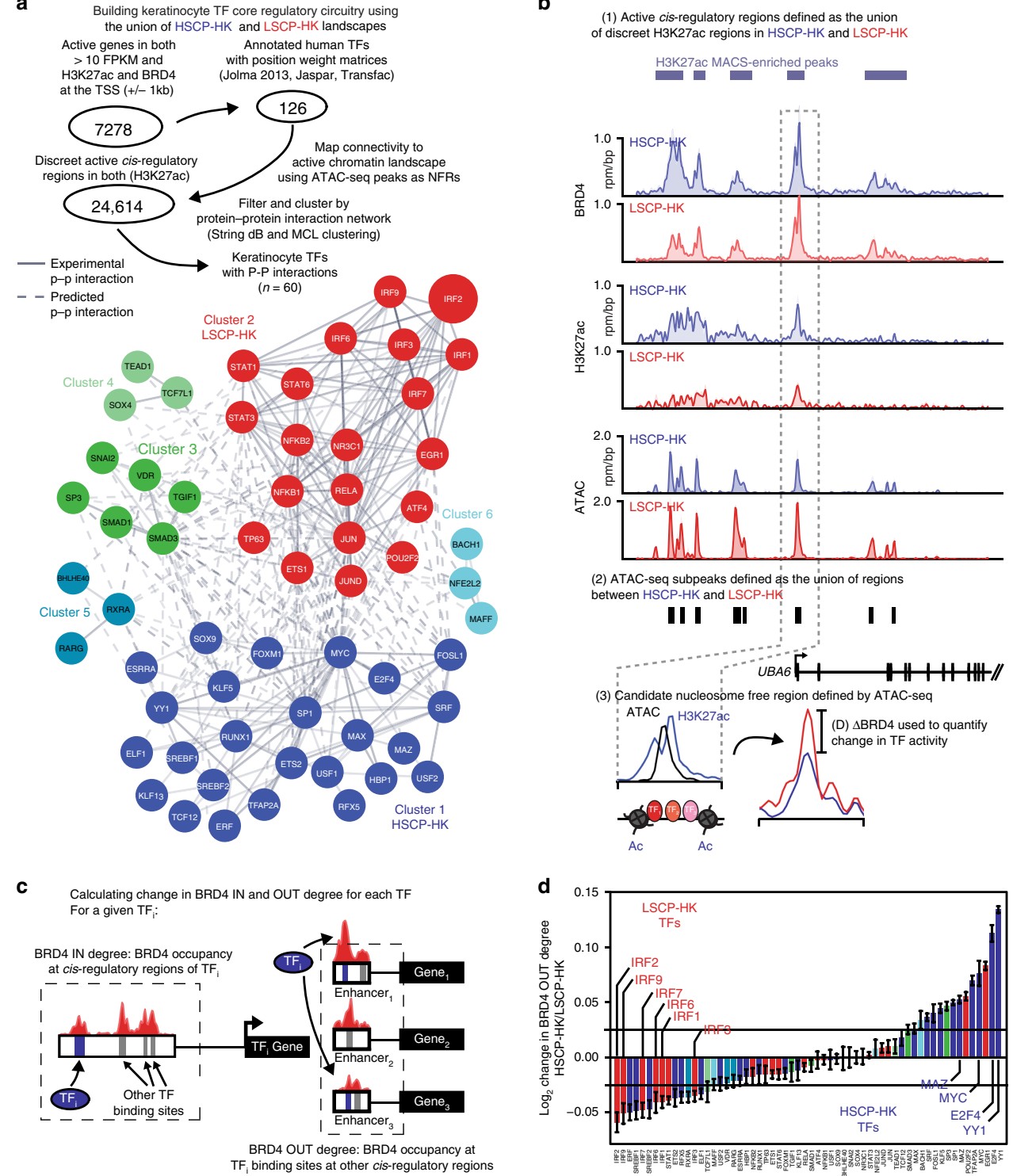

**Fig. 3** Changes in TF core regulatory circuitry between HSCP-HKs and LSCP-HKs. **a** Schematic showing method to identify potential CRC TFs. The union of active genes and H3K27ac sites in both HSCP-HKs and LSCP-HKs was formed in order to create the overall active *cis*-regulatory landscape for HKs. Subsequently, trans factors and *cis*-regulatory elements within the HK regulatory landscape showing LSCP or HSCP specific activity were identified, resulting in network of 60 candidate CRC TFs. Each TF node is colored based on Markov Chain Linkage clustering. Solid and dashed edges represent experimental and predicted protein–protein interactions, respectively, as determined by the StringDB protein–protein interaction database. **b** Tracks of BRD4 and H3K27ac ChIP-seq as well as chromatin accessibility (ATAC-seq) signal at UBA6 genomic locus with schematic detailing how ATAC-seq peaks are used for TF motif searching followed by quantification of proximal changes in BRD4. **c** Schematic showing how BRD4 IN and OUT degree are calculated for each CRC TF. **d** Ranking of the fold change in BRD4 OUT degree for all CRC TFs in HSCP-HK over LSCP-HK. Error bars represent 95% CI of the mean as determined by resampling with replacement (1000 permutations)

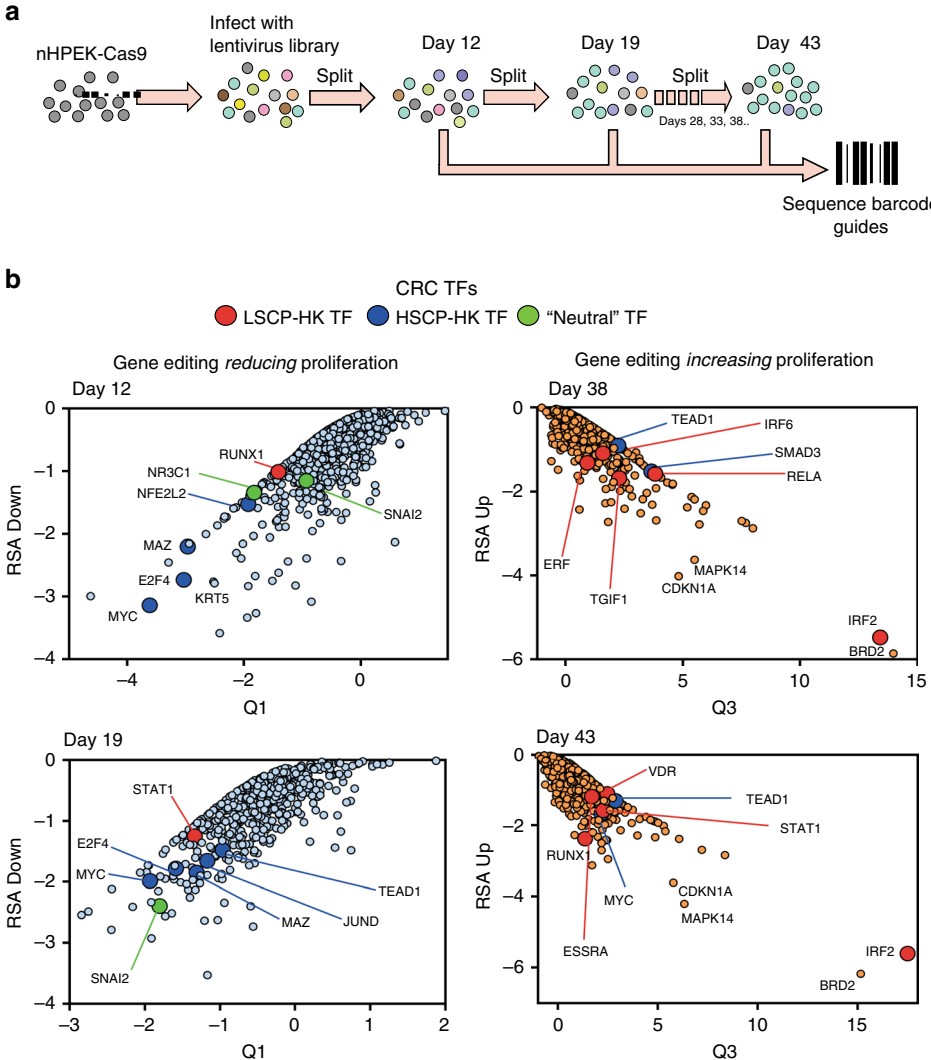

**Fig. 4** CRISPR–Cas9 screen confirms CRC-TFs' role in keratinocyte stem cell fate. **a** Schematic depicting pooled screening approach in Cas9-expressing neonatal HPEKs (nHPEK-Cas9) infected with sgRNA library (at day 0) followed by puromycin selection and collection of cell pellets at days 12, 19, 28, 33, 38 and 43 post infection after split. Abundance of sgRNA sequences was quantified by deep sequencing of the corresponding barcodes in the genomic DNA of the different cell pellets. **b** Gene level results of pooled screen at days 12, 19, 38, and 43, plotting z-score (Q1 or Q3) versus redundant siRNA activity (RSA Down or RSA Up). Q1 versus RSA Down at Days 12 and 19 identifies gene editing that reduces proliferation while Q3 versus RSA Up at Days 38 and 43 identifies gene editing that increases proliferation. The top 7 Core Regulatory Circuitry (CRC) TFs (from Fig. 3d) are highlighted and marked as associated with either HSCP (blue), LSCP (red), or "neutral" (green)

onwards (Supplementary Fig. 5e) as IRF2 was predicted as the top CRC-TF regulating LSCP-HKs (Fig. 3d). IRF2 knockdown (KD) in mice leads to epidermal hyperplasia suggesting a physiological role for IRF2 in promoting a more differentiated, less proliferative epidermis[48]. Furthermore, IRF6 and RELA have been shown to negatively regulate keratinocyte proliferation[49,50]. Besides TFs, KRT5 editing was shown to quickly reduce keratinocyte proliferation, confirming its essential role in basal keratinocytes[51] (Fig. 4b). CDKN1A (p21) and MAPK14 (p38α) editing was shown to increase cell proliferation and could have been counteracting senescence associated cell cycle arrest[52,53] possibly related to DNA damage induced by CRISPR–Cas9[54] (Fig. 4b, days 38 and 43). In addition to IRF2, BET bromodomain protein BRD2 knockout was strongly associated with enhanced cell proliferation (Fig. 4b, Supplementary Fig. 5e).

Some top CRC-predicted TF regulators did not validate in the screen, possibly due to technical limitations and variability in CRISPR editing efficiency or assumptions around TF occupancy

used for the CRC analysis. For example, only 3 of the 5 sgRNA guides added against our top hits BRD2 and IRF2 remained at the end of the screen (Supplementary Fig. 5e), despite four IRF2 and five BRD2 sgRNAs showing efficient editing (Supplementary Fig. 6a). IRF2 sgRNAs (#2,#3,#4) enhanced keratinocyte migration while sgRNAs #1, #2, #3, and #4 enhanced clonogenicity, confirming IRF2 as a negative regulator of stem cell function (Supplementary Fig. 6b, c). Interestingly, although all BRD2 sgRNA enhanced the clonogenic potential of keratinocytes (Supplementary Fig. 6c) no effect on migration was observed (Supplementary Fig. 6b). One explanation could be a time dependent increase in stem cell function only captured in the longer clonogenic assay. The BRD2 findings are interesting in light of recent studies describing its role in keratinocyte inflammatory responses[55] and warrants further research. However, for this study we decided to focus on IRF2 as its KD led to robust and reproducible phenotypic changes in keratinocytes.

**CRISPR–Cas9 editing of TFs modulates stem cell function.** To orthogonally validate the role of select TF candidates, we repeated genetic loss of function studies in adult keratinocytes. Editing of YY1 and SNAI2 in HSCP-HKs resulted in a loss of clonogenicity and migration (Fig. 5a). Further physiological relevance was evaluated in a 3D human dermo-epidermal skin model that tests the ability of keratinocytes to form a stratified epidermis when combined with an artificially constructed dermal layer[56], a model previously shown to link keratinocyte stemness, delayed replicative senescence and increased epidermal thickness[44]. YY1 KD and SNAI2 KD in adult keratinocytes significantly impaired the formation of an epidermis and failed to maintain an undifferentiated basal layer (Fig. 5b, Supplementary Fig. 7a, b). Furthemore, YY1-KD and SNAI2-KD keratinocytes failed to migrate under the dermis as seen with control keratinocytes (Fig. 5b, Supplementary Fig. 7a, b), confirming both YY1 and SNAI2 as master regulators of keratinocyte stem cell function. To determine whether either YY1 or SNAI2 are suffcent to induce keratinocyte stem cell function would require their individual overexpression but this was not performed in this study.

IRF2 edited keratinocytes took longer to display phenotypic changes and were passaged until their proliferation rate surpassed controls. IRF2-KD keratinocytes showed increased clonogenicity and migration (Fig. 5c) and formed a thicker, more cellular stratified epidermis in the 3D human skin model, demonstrating both a retention of keratinocyte identity and greater stem cell potential (Fig. 5d). Furthermore, IRF2-KD cells were highly migratory generating an epidermis beneath the dermal construct which prevented statistical analysis of epidermal thickness (Fig. 5d, Supplementary Fig. 7d). To further characterize IRF2-KD keratinocytes in the clonogenic assay, we quantified Holoclones (formed by cells with highest proliferation and lifespan), Paraclones (formed by cells with a limited lifespan) and Meroclones (a transitional stage between between holoclones and paraclones)[57]. IRF2 KD keratinocytes form more holoclones (Fig. 5e), reflecting greater stem cell function and self-renewal capacity[9].

To evaluate IRF2 specificity, we edited another family member, IRF9, in HSCP-HKs. IRF9-KD cells showed no increased clonogenicity or migration (Supplementary Fig. 7c) or epidermal thickness in the 3D model but a better delineation of the undifferentiated basal cell layer (Supplementary Fig. 7a, b).

**IRF2 knockout restores HSCP-HK gene-expression programs.** As IRF2 KD in HSCP-HK cells prevented loss of stem cell function over time in vitro, we next asked whether IRF2 KD in LSCP-HK cells could restore stem cell function and transcriptional signature of HSCP-HKs (also edited for comparison). IRF2 KD induced global transcriptional changes in LSCP-HKs toward those of HSCP-HKs (Supplementary Fig. 8a–c), illustrated using genome-wide clustering (Fig. 6a). Cluster 1 showed genes upregulated in LSCP-HKs, compared to HSCP-HKs, and downregulated by IRF2 KD toward HSCP-HKs expression. Cluster 2 showed genes downregulated in LSCP-HKs, compared to HSCP-HKs, and upregulated by IRF2 KD toward HSCP-HKs expression. Both clusters represent over 70% of genes that are reversed by IRF2 KD in LSCP-HKs toward HSCP-HK expression levels (Fig. 6a).

To profile IRF2-binding landscape and effects on gene expression, we created keratinocytes with a doxycycline-inducible allele of HA-tagged IRF2 on a CRISPR–Cas9 generated IRF2 knockout background (Supplementary Fig. 9a) to perform anti-HA-tag ChIP-Seq. We identified 2068 high confidence sites of IRF2 binding, using an anti-HA pulldown in nonengineered HSCP-HKs as an additional control for spurious ChIP-seq peaks.

Half of all IRF2 binding sites occurred at genomic regions with chromatin accessibility (ATAC-seq) and half at regions without. Both sets of regions were strongly enriched for the IRF2 motif (Supplementary Fig. 9b, c). As non-ATAC IRF2 sites occurred far from genes and were not accounted for in our prior transcriptional circuitry analysis, we focused subsequent analysis on the half of IRF2 binding sites co-localizing with ATAC-seq peaks.

Integration of IRF2 binding with clustering of IRF2 KD gene expression showed a significant relationship between genes with direct IRF2 binding and those elevated in LSCP-HKs versus HSCP-HKs and downregulated by IRF2 KD (Fig. 6a, Cluster 1). Genes directly bound by IRF2 included those associated with interferon response and MHC Class I antigen presentation such as β-2-microglobulin (B2M), HLA-C, and immunoproteasome proteins (PSMB8, 9, 10). Figure 6b illustrates IRF2 binding at the bidirectional promoter region of TAP1/PSBM9, with evidence of strong direct regulation of PSMB9 (Fig. 6c). Overall, predicted IRF2 binding sites that were validated by IRF2-HA ChIP-seq showed a more pronounced effect in comparing BRD4 occupancy between HSCP-HKs and LSCP-HKs (Fig. 6d). In general, the top 100 IRF2 target genes showed lower expression in response to IRF2 KD or in HSCP-HKs versus LSCP-HKs (Fig. 6e, Supplementary Fig. 9d). Finally, ranking of genes by proximal IRF2 occupancy showed strong leading edge enrichment for interferon response gene expression programs (Fig. 6f), recapitulated in functional enrichment analysis of the top 100 IRF2 target genes (Fig. 6g). These data suggest that IRF2 is a direct activator of interferon response and antigen presentation in LSCP-HKs.

Further exploration of gene expression in IRF2-KD keratinocytes revealed additional changes in genes not directly bound by IRF2. Cluster 1 showed such genes upregulated in LSCP-HKs compared to HSCP-HKs and downregulated by IRF2 KD toward the expression in HSCP-HKs. The top GO terms for Cluster 1 are "Keratinization" and "Cornification", processes involved in terminal differentiation, cell death and the formation of a cross-linked insoluble barrier, suggesting that IRF2 editing is associated with keratinocyte differentiation (Supplementary Fig. 9e, Supplementary Fig. 10). Two important genes in these processes, involucrin and SPRR1A (small proline rich protein 1A), components of the insoluble cross-linked envelope critical for skin barrier formation, were reduced by IRF2 KD in LSCP-HKs back to HSCP-HK levels Supplementary Fig. 9e). Interestingly, most of the genes in the subcluster 3, which were reversed back to HSCP-HKs (28 genes), are involved in GO term "Keratinocyte Differentiation" (Supplementary Fig. 10).

Cluster 2 is the largest cluster and characterized by genes downregulated in LSCP-HKs compared to HSCP-HKs and upregulated by IRF2 KD in LSCP-HK to the levels of HSCP-HKs (Fig. 6a). The top GO term associated with this cluster is "Cell Cycle", as cell cycle-associated genes were restored to HSCP-HK levels after IRF2 editing in LSCP-HKs, suggesting IRF2 reduces cell proliferation after serial passaging (Fig. 6a, Supplementary Fig. 9e). These genes were also increased in HSCP-HKs with IRF2 KD, suggesting that cell cycle potential can be further increased even in low passage keratinocytes. Further sub-clustering showed that expression of most cell cycle associated genes was upregulated towards HSCP-HK levels following IRF2 KD in LSCP-HKs (Supplementary Fig. 11). Interestingly, CDKN2A (p16INK4a) was the only cell cycle associated gene linked with cluster 1, hence upregulated in LSCP-HKs and restored to HSCP-HK levels with IRF2 editing (Fig. 6a), supporting a reversal of senescence-associated cell cycle arrest.

Cluster 3 showed genes largely unchanged by IRF2 editing in both LSCP- and HSCP-HKs. Cluster 4 showed genes upregulated by IRF2 KD both in LSCP- and HSCP-HKs.

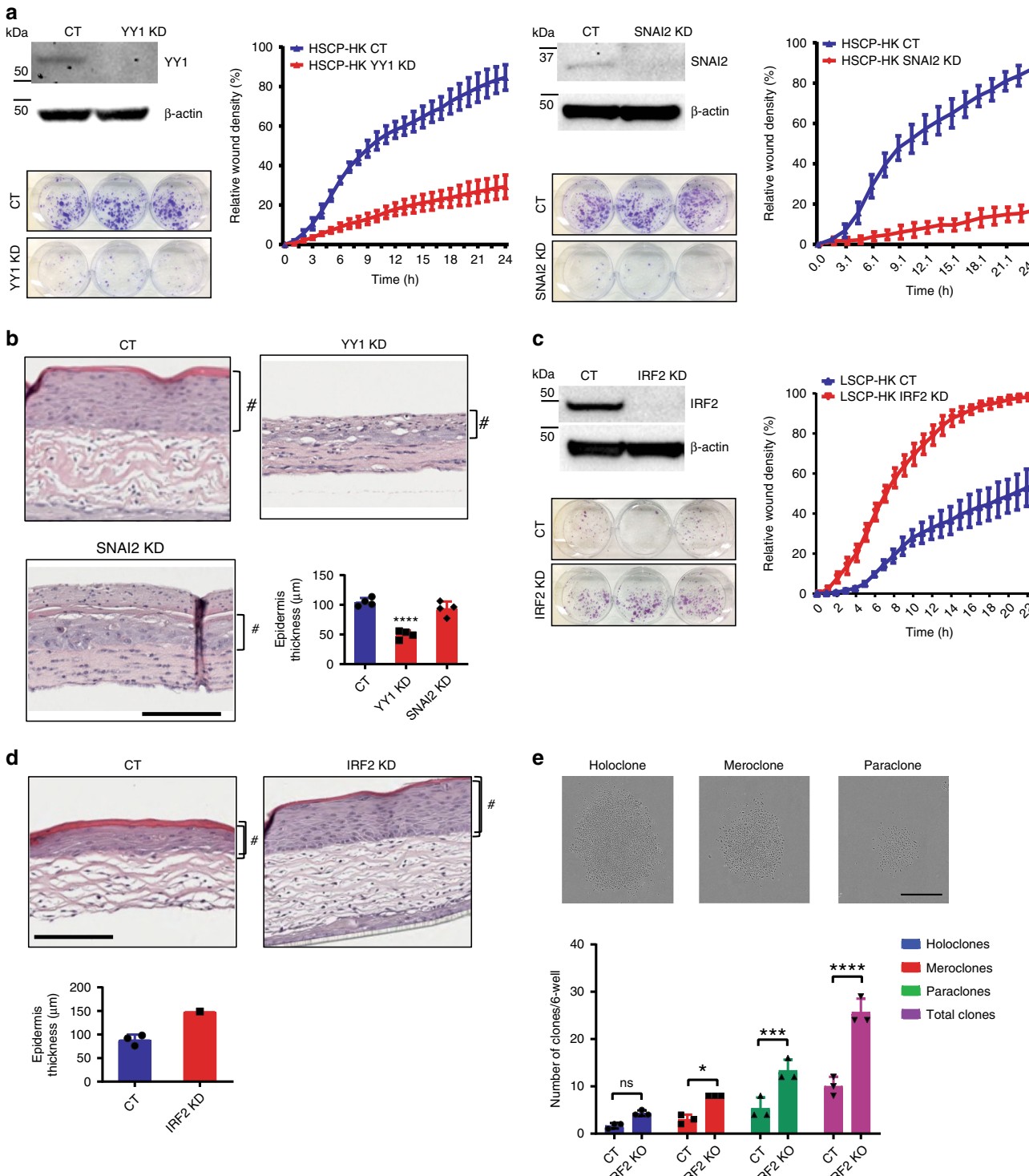

**Fig. 5** Loss of IRF2 induces stem cell potential whereas loss of YY1 or SNAI2 reduces it. **a** Assessment of CRISPR–Cas9 mediated knockdown of YY1 (left panel) or SNAI2 (right panel) on colony formation and migration (as in Fig. 1) in HSCP-HK compared to control cells, with protein levels monitored by Western blot analysis utilizing β-actin protein level as a loading control. **b** Comparison of SNAI2 KD and YY1 KD to control cells in their functional ability to generate an epidermis in human dermo-epidermal 3D model. # indicates Epidermis layer. Epidermis thickness (μm) was quantified for each condition and differences assessed using One-way ANOVA with Holm–Šídák multiple comparisons test. ****$p < 0.0001$. Scale bar = 167 μm. Means ± S.D. of $n = 4$ biological replicates. **c** Assessment of colony formation and migration in LSCP-HK cells with CRISPR–Cas9 mediated knockdown of IRF2 compared to control cells, with protein levels monitored by Western blot analysis utilizing β-actin protein level as a loading control. **d** Epidermis formation in human dermo-epidermal 3D model comparing IRF2 KD cells to controls. # indicates epidermis layer. Epidermis thickness (μm) was quantified for each condition (means ± S.D. of $n = 3$ biological replicates for CT and $n = 1$ for IRF2 KD). Scale bar = 167 μm. **e** Comparison of ability to clonally expand aHPEK-IRF2-KD versus control cells, where individual keratinocytes either fail to form a colony or form a colony belonging to three broad morphological types termed holo-, mero-, and paraclones (pictured, scale bar = 1.1 mm). Total colonies of each morphological type were counted and plotted as means ± S.D. of $n = 4$ biological replicates. *$p < 0.05$, ***$p < 0.001$, ****$p < 0.0001$ (Ordinary One-way ANOVA with Holm–Šídák multiple comparisons test)

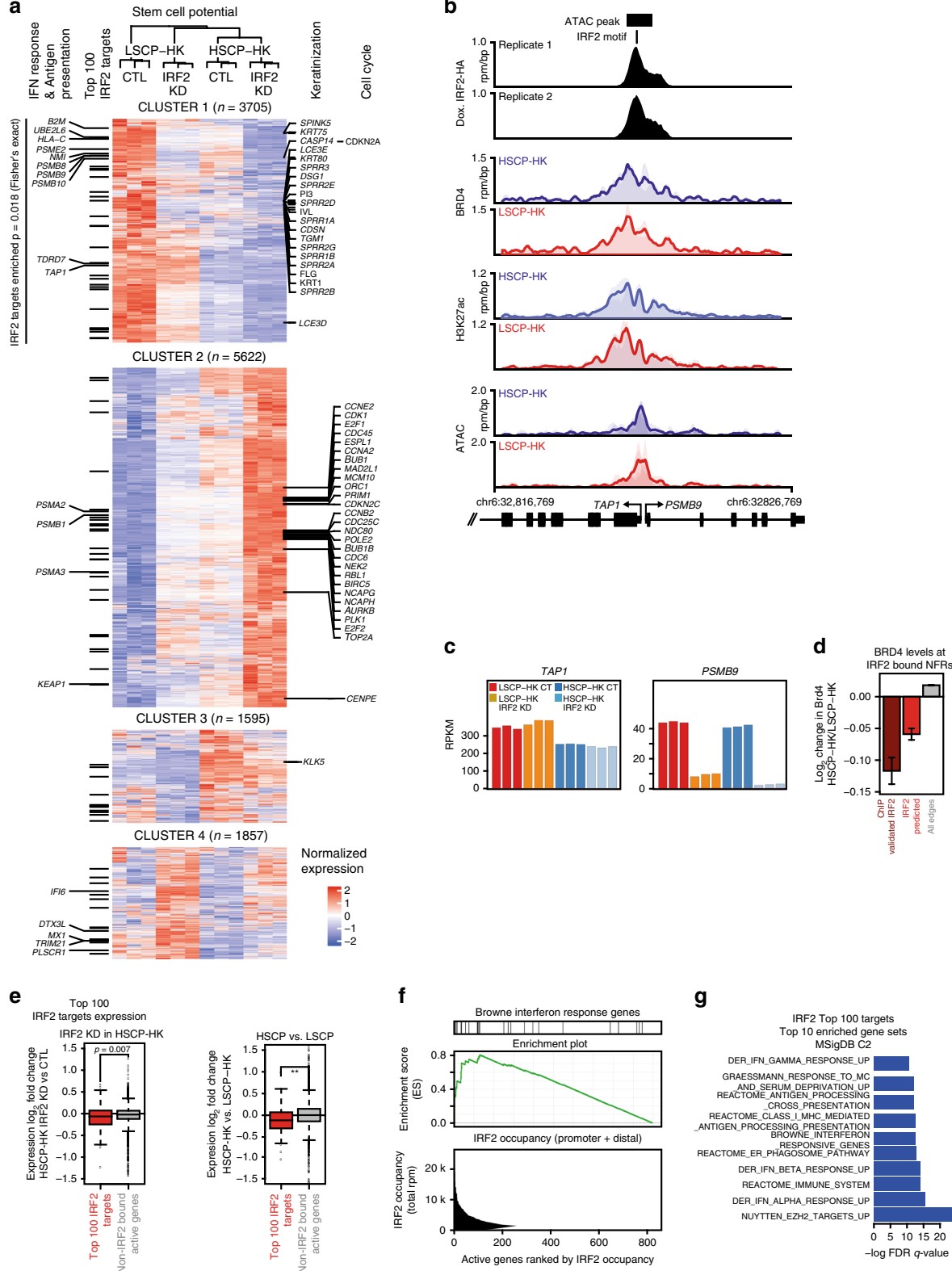

IRF2 is best known as an antagonist to interferon mediated antiviral gene activation by IRF1[58] but is also associated with gene activation[59], as recapitulated in our data. GO term "Interferon Signaling" was strongly associated with LSCP-HKs as compared to HSCP-HKs, implicating interferon signaling in stem cell function (Supplementary Fig. 12). However, IRF2 KD in LSPC-HKs only partially reduced the genes ascribed to this GO term

back to HSCP-HK levels, implying that interferon signaling is not the dominant driver of loss of of stem cell potential (Supplementary Fig. 12). Inflammatory response-related genes are strongly associated with LSCP-HKs as compared to HSCP-HKs and also with LSCP-HKs CT versus LSCP-HKs IRF2 KD (Supplementary Fig. 13). Several of these genes were upregulated and reversed by IRF2 KD to HSCP-HK levels such as IL1B,

**Fig. 6** IRF2 loss of function restores HSCP-HK gene expression programs. **a** Hierarchical clustering (*k*-means = 4) of global gene expression (RNA-seq) contrasting the effect of IRF2 KD in LSCP-HK or HSCP-HK versus respective control cells. Color-code highlights gain (red) and loss (blue) across conditions. Top 100 IRF2 target genes as well as those IRF2 target genes associated with interferon response or antigen presentation are indicated on the left. Genes associated with top Gene Ontology (GO) terms from the LSCP-HK IRF2 KD versus CT expression analysis are listed on the right of the heat map (Keratinization and Cell Cycle, showing only genes with absolute log2FC > 1). **b** Tracks of IRF2-HA, BRD4, and H3K27ac chromatin occupancy (ChIP-seq) as well as chromatin accessibility (ATAC-seq) signal at the *TAP1/PSMB9* genomic locus. **c** Transcript levels based on RNA-seq (RPKM) of TAP1 and PSBM9 upon IRF2 KD in either LSCP-HK or HSCP-HK across individual replicates. **d** Average change in BRD4 OUT degree at IRF2 edges that are either IRF2 ChIP validated (left, dark red), predicted (center, red) versus all other edges (right, grey). Error bars represent standard error of the mean. **e** Expression changes for top 100 IRF2 target genes versus other non-IRF2 bound active genes upon IRF2 KD in HSCP-HK (left) or when comparing HSCP-HK versus LSCP-HK (right). Significance of the difference in the distributions is denoted by a two-tailed *t* test **$p < 1e-6$. Box represents the 25–75 percentile, line represents the median, and whiskers extend 1.5× the 25–75 percentile range. **f** Ranking of the 855 genes with detectable IRF2 binding by total proximal IRF2 binding (from left to right, bottom panel), plotting cumulative IRF2 signal at each gene measured in units of total reads per million (rpm) of IRF2-HA ChIP-seq signal at the promoter and proximal enhancers. Top panel annotates genes associated with interferon response, middle panel shows leading edge enrichment of this gene set (GSEA). **g** Ranking by FDR *q*-value of the top 10 most highly enriched gene sets (MSigDB C2) associated with top 100 IRF2 target genes

S100A8, and PTGS2 (Supplementary Fig. 13). Although TNF was increased in LSCP-HKs, IRF2 KD had no effect on TNF gene expression and only partially inhibited CXCL8. As shown in Supplementary Fig. 1c, various senescence-associated genes were either upregulated or downregulated in LSCP-HK compared to HSCP-HK as anticipated from their enhanced SA-Gal staining. IRF2 KD restored the expression of many of these genes, including AURKA, AURKB, CCNA2, CCNB1, CDKN2A (p16INK4), FOXM1, IL1B, MMP3, PCNA, PTGS2, and UBE2C, strongly suggesting a function as a positive regulator of keratinocyte senescence.

Our data showed a strong association between genes upregulated in LSCP-HK cells (compared with HSCP-HKs) and those upregulated in psoriatic compared to normal skin (Supplementary Fig. 14). Furthermore, we detected an equally strong association when comparing differential expression between LSCP-HKs CT and LSCP-HKs IRF2 KD with genes expressed in psoriatic skin, suggesting that many psoriasis-related genes were upregulated by serial passaging and reversed by IRF2 KD to low passage-HSCP-HK levels (Supplementary Fig. 13).

As mentioned before, IRF2 can act both as an inhibitor and activator of gene expression[58,59]. IRF2 binding was predicted to be increased in the regulatory regions of various genes (Fig. 3d, Supplementary Data 4) in LSCP-HKs compared to HSCP-HKs, and among these putative target genes we found expression to be both upregulated and downregulated when comparing LSCP-HK and HSCP-HK (Supplementary Fig. 15 left), arguing that an increased binding of IRF2 either activated or inhibited expression of a specific gene (Supplementary Fig. 15 left). The resulting genome-wide hierarchical clustering revealed two large clusters (2 and 3) and two smaller ones (1 and 4) where genes in cluster 2 tended to be upregulated in LSCP-HK but reduced by IRF2 KD. Based on this clustering, our prediction would be that PTGS2, NFKBIZ, and SOX4 are directly upregulated by IRF2. Similarly, cluster 3 genes tended to be downregulated in LSCP-HKs and increased by IRF2 KD. In this case, our prediction would be that AURKA and AURKB are directly inhibited by IRF2 thus linking IRF2 to senescence-related genes (Supplementary Fig. 15 right).

**CRISPR-KO of IRF2 in LSCP-HKs restores stem cell function.** Having established that IRF2 KD in LSPC-HKs reverses the genome-wide gene signature toward that of HSCP-HKs, we evaluated its effect on stem cell function. IRF2 KD by CRISPR–Cas9 editing was confirmed in HSCP-HKs and LSCP-HKs by Western blot (Fig. 7a). The senescence-associated protein p16 showed significantly higher expression in LSCP-HKs than HSCP-HKs as expected (Fig. 7a) and IRF2 KD significantly reduced p16 expression in both populations (Fig. 7a) supporting a

reversal of the senescence phenotype suggested by transcriptional profiling. IRF2 editing in LSCP-HK restored clonogenicity towards a HSCP phenotype (Fig. 7a, b), indicating a functional effect of the induction of cell cycle genes (Fig. 6). The migration assay also showed a restoration of HSCP-HK phenotype in IRF2 KD LSCP-HK and a further enhanced migration potential in IRF2 KD HSCP-HK over control cells (Fig. 7c). Interestingly, β-galactosidase staining was also reduced in LSCP-HKs by IRF2 editing, again suggesting a role as a TF involved in cellular senescence (Fig. 7d) (supporting data from Fig. 6, Supplementary Fig. 11). IRF2 KD in LSCP-HK cells also restored the ability to form a thicker and more cellular epidermis in the 3D human skin model (Fig. 7e, Supplementary Fig. 16a, b). Again HSCP-HKs establish a clear undifferentiated basal layer (Fig. 7f) which is less convincing using LSCP-HKs where this layer is disorganized with few cells (Fig. 7f). IRF2 editing of LSCP-HKs results in less differentiation, a thicker epidermis and increased cellularity in the basal layer of 3D skin models (Fig. 7f, Supplementary Fig. 16a, b).

## Discussion

While epithelial stem cell biology has advanced significantly over the last few decades, key questions remain as to the nature, plasticity and regulation of adult epithelial stem cells. Further insights are needed to guide successful therapeutic intervention to prevent epithelial damage or promote epithelial regeneration in disease or ageing. This study delineates the transcriptional circuitry imposing epidermal stem cell function, demonstrating that transcriptional regulatory circuits, previously shown to regulate cell identity, also govern the more nuanced phenotypic differences between human keratinocytes with low and high stem cell potential.

Using an unbiased approach based on stem cell function rather than marker expression, we exploited the loss of epidermal stem cell function over time in vitro for a high yield of keratinocytes with differential stem cell function. Confirming that replicative senescence paralleled increases in differentiation markers in our model, we employed transcriptional circuitry analysis to identify TFs predicted to modulate stem cell function, followed by validation thereof using a CRISPR-Cas9 screen. The validity of our screening approach was demonstrated with the inclusion of guides to known TF regulators of stem cell fate such as SNAI2[60], which were rapidly depleted from the pool. In addition, we successfully identified a number of TF regulators of epidermal stem cell fate. An outstanding and unexpected finding from this screen was the persistence and (very) strong enrichment of IRF2 edited cells in the final pool, confirming the utility of our transcriptional circuitry analysis and suggesting that IRF2 was significantly antagonistic to epidermal stem cell function.

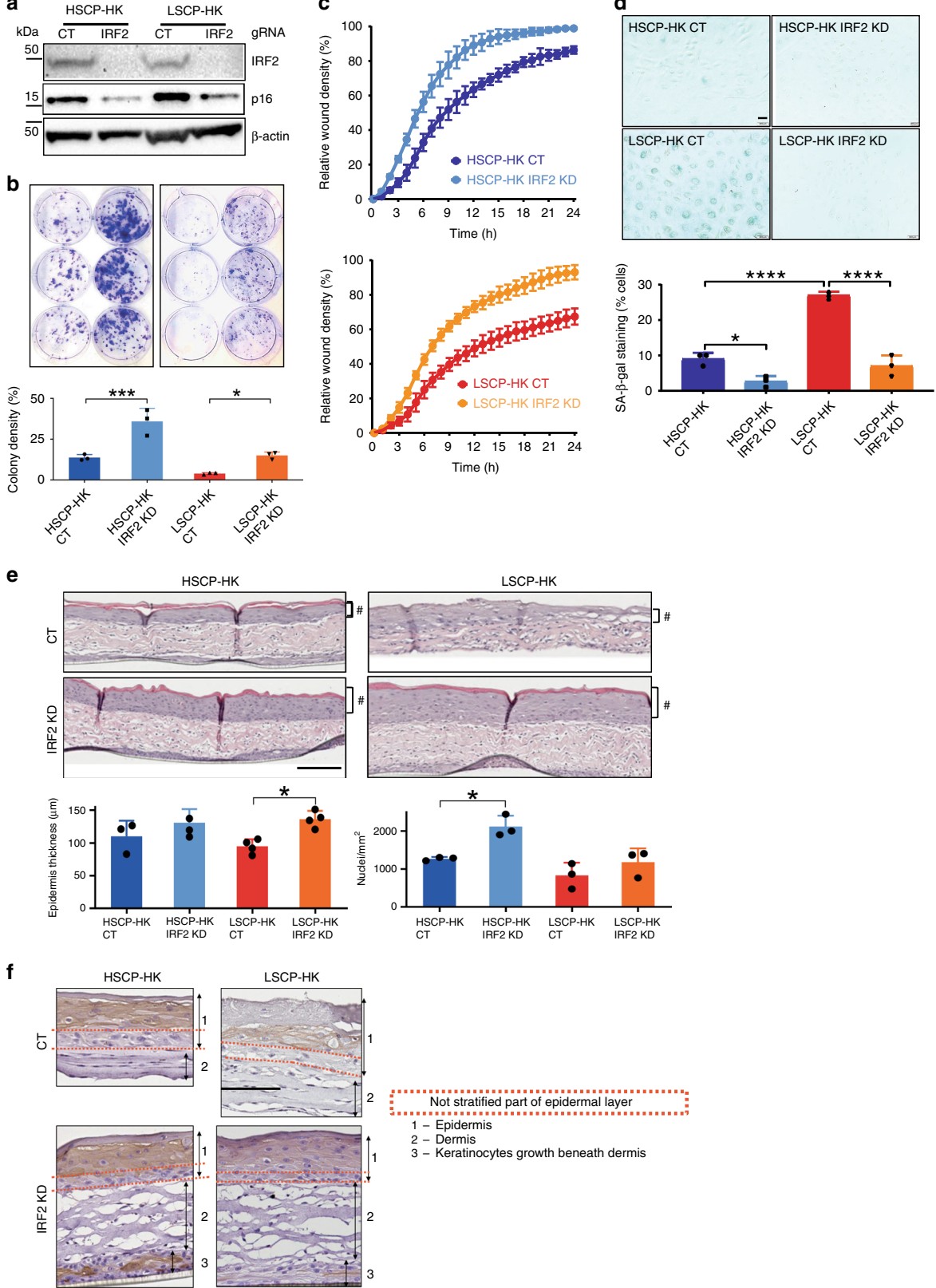

We demonstrate that IRF2 is antagonistic to adult human epidermal stem cell function as IRF2 edited keratinocytes exhibited enhanced clonogenicity[8] and migratory characteristics[23]. Importantly, IRF2 KD cells retained their keratinocyte identity demonstrated by their formation of a stratified squamous epidermis in a human 3D skin model which was thicker and less differentiated than control cells[44]. Furthermore, IRF2 editing could restore stem cell function and proliferative capacity in high passage keratinocytes that had undergone replicative senescence. These findings support a major role for IRF2 in driving loss of keratinocyte stem cell function although this was not directly demonstrated with overexpression experiments. Interestingly,

**Fig. 7** IRF2 loss of function in LPSC-HK restores stem cell potential. **a** Effect of CRISPR–Cas9 KD of IRF2 in HSCP-HK and LSCP-HK cells on IRF2 and p16 INK4A protein expression as measured by Western blot using β-actin as a loading control. **b** Clonogenic assay comparing IRF2 KD in HSCP-HKs (left panel) versus LSCP-HK (right panel) and quantification of colony forming potential as a function of cell density. Colony density was measured in % area (means ± S.D. of $n = 3$ biological replicates). $^{*}p < 0.05$, $^{***}p < 0.001$ (One-way ANOVA with Holm–Šídák multiple comparisons test). **c** Keratinocyte migration assay comparing IRF2 KD in HSCP-HK (top panel) versus LSCP-HK (bottom panel). **d** Beta-galactosidase (SA-β-gal) staining as a marker of senescence upon IRF2 KD in HSCP-HK versus LSCP-HK cells. SA-β-gal staining was quantified as % of cells with positive staining (means ± S.D. of $n = 3$ biological replicates). $^{*}p < 0.05$, $^{****}p < 0.0001$. (One-way ANOVA with Holm–Šídák multiple comparisons test). Scale bar = 200 μm. **e** Quantification of epidermis formation upon IRF2 KD in HSCP-HK cells versus LSCP-HK cells. * indicates $p < 0.05$ (One-way ANOVA with Holm–Šídák multiple comparisons test), # indicates the epidermal layer. Epidermis thickness (μm) was measured for each condition at 3 sites within the construct (means ± S.D. of $n = 4$ biological replicates). The number of DAPI-stained nuclei were counted per mm² for each condition. Scale bar = 167 μm. **f** Immunocytochemical staining for Keratin 10 in human dermo-epidermal 3D model comparing IRF2 KD in HSCP-HK versus LSCP-HK. Keratin 10 is used as an epidermal differentiation marker expressed only by keratinocytes in suprabasal epidermal layers in human skin. Keratin 10 staining reflects differentiation patterns within 3D skin constructs where the presence and structure of a basal (stem cell) layer does not stain. For clarity, epidermis is labeled as layer 1, dermis as layer 2 and keratinocytes growing beneath the dermis are labeled as layer 3. Red dotted lines illustrate the extent of the unstained basal layer within the epidermis. While all human skin constructs maintain an unstained (undifferentiated) basal layer, differences between conditions are apparent. Scale bar = 128 μm

IRF2 expression showed only minimal change upon serial passaging of cells, which suggests that posttranscriptional regulation of IRF2 activity or differential interaction with other proteins might drive IRF2's chromatin-directed functions.

IRF2 has been broadly associated with inflammatory diseases of the skin including psoriasis, but the relative contributions of epithelial and immune cells are difficult to discern with IRF2 KO models. IRF2 regulates pro-inflammatory response in macrophages[61] and data[48] suggest that IRF2 could also impair epidermal function critically required to maintain the essential skin barrier. Of note, a recent analysis of adult and fetal proerythroblast cells showed enrichment of IRF2 at adult-specific active promoters of genes associated with type I interferon and interferon gamma signaling[62]. This implies that IRF2 may have a broader role in "ageing" and specifying cell state between fetal and adult cells. Our data suggest that IRF2 drives type I interferon and gamma-interferon signaling in keratinocytes but the extent to which these pathways drive the loss of stem cell function requires further study. However, there is a growing appreciation of a role for inflammation in regulating stem cell behavior, most notably described in HSCs which are driven out of quiescence by interferon signaling[63], but also observed in intestinal stem cell activation[64]. Importantly, low level systemic inflammation is associated with chronic diseases of ageing and may play a role in causation[65], supporting the importance of further mechanistic studies to define the role of IRF2 in epithelial biology.

There is considerable evidence that a cell's gene-expression program is controlled by a limited and cell-specific number of TFs[66]. Such core TFs are found enriched at super-enhancers, large clusters of transcriptional enhancers, and they form interconnected, auto regulatory and feedforward circuits to impose cell identity[15,21] TF regulatory circuits can induce pluripotency[14] and transdifferentiate cell types[67]. Indeed, a recent study shows in vivo transdifferentiation of mesenchymal cells[68] into keratinocytes using TP63, GRHL2, TFAP2A and MYC to enable healing of large skin wounds[68]. Such studies underscore the validity of TF-driven therapeutic approaches and interestingly two of the TFs used to convert mesenchymal to epithelial skin lineage, TFAP2A and MYC, were in our leading edge for TFs associated with high stem cell potential.

This study adds to a growing understanding of the importance of TF networks in health and disease. Our findings reveal that, in addition to specifying cell identity, transcriptional networks define cell state within a single lineage thereby adding an additional layer of "fine-tuned" regulation of cellular plasticity. Refinement of the computational and chromatin profiling technology used in this study, particularly applied to low cell numbers, holds promise for the discovery of significant targets for disease-modifying small molecule therapeutics. Once considered "undruggable", TFs are now an intense focus of chemistry efforts to enable small molecule regulation of their activity for therapeutic benefit[69,70]. Although the current focus for such efforts is cancer, the ability to identify dominant TF regulators of cell state broadens potential therapeutic utility. Small molecules against key cell fate regulators could enable manipulation of cell plasticity for therapeutic benefit in tissue engineering, regenerative medicine and other disease settings.

## Methods

**Human keratinocyte culture and serial passaging.** All human cells were purchased from approved commercial suppliers who had obtained the necessary patient consent for skin donation for research purposes. The cells were obtained and handled in compliance with the Swiss Human Research Act. Normal human neonatal epidermal keratinocyte progenitor cells (nHPEKs, Lot#EB1110044 (pooled)) and adult normal human epidermal keratinocyte progenitor cells (aHPEKs, Lots #ES1303277, and #MC1511045 (both abdominal)) were purchased from CELLnTEC (Bern, Switzerland) at Passage 2 (P2). HPEKs were expanded until Passage 4 (P4) and pellets of $1 \times 10^6$ cells were cryopreserved in liquid nitrogen for later use. HPEKs were cultured from P4 by seeding at an initial density of $5.3 \times 10^4$ cells/cm² using CnT-Prime media (CELLnTEC, cat#CnT-PR). When cells reached 70–85% confluency, they were detached from the cell culture dish using Accutase Cell Detachment Solution (CELLnTEC). For serial passaging experiments, aHPEKs were grown and passaged about 13 times. aHPEKs were collected for various assays and were considered as "HSCP-HKs" when fast-dividing (between Passages 7–9) whereas aHPEKs that were slow-dividing (Passages 15–17) were considered as "LSCP-HKs".

**Clonogenic assay.** HPEKs were seeded at 1000–2000 cells/well in 6-well plates (in triplicates) with 5 ml of EpiLife Medium (ThermoFisher Scientific, Zug, Switzerland) with no BPE and 75–85% of human keratinocyte growth supplements (HKGS, ThermoFisher Scientific). 80% of the medium was replaced every 3 or 4 days. After 8–12 days plates were washed with phosphate-buffered saline (PBS) and colonies fixed with 10% neutral buffered formalin solution for 20 min. After another wash with PBS, colonies were stained with 0.5% (w/v) crystal violet (50% of 1% crystal violet and 50% Ethanol) for 45 min. Excess crystal violet was removed by washing with water and then dishes were dried. Images of stained colonies were taken using a white background. Cell confluency was measured with the IncuCyte ZOOM™ Live-Cell Imaging System and analyzed with the Basic Analyzer (Essen Instruments, Ann Arbor, MI, USA) by whole-well imaging of the stained wells.

**ECM-binding assay.** In total, 96-well NUNC plates were coated with different ECM for two days at 4 °C; 25 μg ml⁻¹ Collagen I (Corning, VWR, Vienna, Austria, cat#354236) (in 0.02 N acetic acid), 25 μg ml⁻¹ Fibronectin (Gibco, cat#PHE0023) (in PBS) or 10 μg ml⁻¹ Laminin V (BioLamina, cat#LN111-02) (in PBS with Ca²⁺/Mg²⁺). Unbound Collagen I and Fibronectin were removed with PBS and excess Laminin V with PBS with Ca²⁺/Mg²⁺. Wells were then blocked with 0.5% heat-inactivated bovine serum albumin (BSA) in PBS (for fibronectin and collagens) or in PBS with Ca²⁺/Mg²⁺ (for Laminin V) and incubated overnight at 4 °C. Totally, $2 \times 10^4$ cells were seeded in to each plate including commercial Collagen IV-coated plates (Corning, cat#08-774-30) and incubated at 15, 30, 45, 60, and 100 min at 37 °C before washing 3 times with PBS. Cells were then incubated with CnT-PR media before applying 15 μM of Resazurin for 75 min. Fluorescence was read at 560ex/590em, sensitivity 40 (Synergy HT from Biotek + Biostack and Gen5 software).

**Migration assay**. HPEKs were detached and resuspended in CnT-FTAL5 medium (CELLnTEC) at $2.0 \times 10^6$ cells/ml. Cells were seeded into ImageLock 96-well Plates (Essen BioScience, Welwyn Garden City, UK, cat#4379) coated with $50\,\mu g\,ml^{-1}$ Collagen I (in 0.02 N) by adding 20 μl cell suspension to 80 μl of Assay Medium (EpiLife Medium supplemented with $30\,\mu g\,ml^{-1}$ BSA (ThermoFisher Scientific, Zug, Switzerland)). After 6 h, the medium was replaced with 90 μl of Assay Medium and cells were starved for an additional 16 h. A wound was introduced into the confluent cell layer by using a 96-pin wound making tool (WoundMaker™, EssenBio). After an additional washing step, 100 μl Assay medium was added and the plates were transferred into the IncuCyte ZOOM (Essen BioScience, cat#9600-0012) for wound area measurement overnight. Wound confluency was monitored and analyzed with the IncuCyte ZOOM™ Live-Cell Imaging System and software according to the supplier's protocols.

**Human dermo-epidermal 3D models**. Human primary dermal fibroblasts (CELLnTEC, Bern, Switzerland, Lot#ES1303222) were expanded in DMEM/F-12 with GlutaMAX™ supplemented with 10% fetal calf serum (ThermoFisher Scientific, Zug, Switzerland) up to passage 6 before used in the dermo-epidermal model. The dermal part was prepared with the RAFT kit (LONZA, Visp, Switzerland, cat#016-0R94) according to the manufacturer's instructions. The cell-collagen solution with a final cell density of $4 \times 10^4$ cells per well was cast into a HTS Transwell®-24-well Permeable Support with 0.4 μm Pore Polyester Membrane and 6.5 mm insert diameter (Corning, cat#3378). RAFT absorbers (LONZA, cat#016-1R33) were placed on top for 15 min to condense the cell–collagen mix. The cells were left to mature for 1 week at 37 °C, 5% $CO_2$ and medium (DMEM/F-12 with GlutaMAX™ supplemented with 5% fetal calf serum and freshly supplemented with 284 μM L-ascorbic acid (Sigma-Aldrich) was changed every 2–3 days. For the epidermal part, CRISPR–Cas9 edited or control edited aHPEKs were seeded on top of the dermal construct. Therefore, medium from the cell–collagen layer was removed from the inserts and the wells and HPEKs, resuspended in CnT-PR-FTAL5 (CELLnTEC), were seeded at a density of $2 \times 10^5$ cells per insert on the top of the cell–collagen layer. CnT-PR-FTAL5 medium (CELLnTEC) was added into the inserts and into the wells, and models were incubated for 3 days whereby HPEKs were covered with medium. The stratification process at the air–liquid interface (ALI) was started by removal of the medium on day 3, leaving the keratinocyte layer exposed to air. Fresh medium was re-applied only to the wells but not to the insert so that the medium level in the former just reached the bottom of the filter insert. The model was kept under ALI conditions for an additional 11 days with a medium change every 2–3 days. Each model was finally fixed with 10% neutral buffered formalin (Sigma), paraffin embedded with the VIP system (Tissue-Tek VIP E300, Sakura). Sections measuring 7 μm thick were prepared and after an automated deparaffinization step the slides were either stained with H&E or with specific antibodies. All stainings were performed by either the Leica STS5020 (Biosystems) for H&E or the Roche DiscoveryXT (VENTANA) using the anti-cytokeratin 10 (Abcam, cat# ab76318, 1/1000 dilution) antibody as a differentiation marker. Slides were scanned and analyzed with the Aperio eSlide Manager (Leica Biosystems) and Leica Web Viewer (Leica Biosystems).

**Beta-galactosidase assay**. HPEKs were seeded between 2.0 and $2.7 \times 10^4$ cells per well in 96-well plates and incubated overnight at 37 °C. β-galactosidase assay was performed using the Senescence beta-galactosidase Kit test (Cell Signaling, Danvers, MA, cat#9860) as per manufacturer's instructions. In brief, cells were initially washed with PBS, fixed and stained with β-galactosidase overnight at 37 °C in a dry incubator. Blue-colored staining for β-galactosidase was captured by taking pictures under a bright field microscope.

**Western blot analysis**. Protein extracts were prepared using RIPA buffer (Sigma: 150 mM NaCl, 1.0% IGEPAL® CA-630, 0.5% sodium deoxycholate, 0.1% sodium dodecyl sulfate (SDS), and 50 mM Tris, pH8.0) completed with Protease Inhibitor (PI) Cocktail tablet (Sigma). Protein concentration was determined with the Pierce BCA protein assay kit (Thermo Scientific). Protein extracts (20-40 μg) were fractionated by SDS polyacrylamide gel electrophoresis on 4–12% Bis–Tris precast polyacrylamide gels (Invitrogen, Paisley, UK) and transferred to a nitrocellulose membrane (Invitrogen). Bands were visualized by chemiluminescence (ECL Plus; GE Healthcare, Hatfield, UK). Protein quantification was performed using ImageJ and statistical significance was assessed using GraphPad Prism 8. Anti-IRF2 (Abcam, cat#ab124744, 1:500 dilution), anti-IRF9 (Cell Signaling, cat#76684, 1:500 dilution), anti-YY1 (Cell Signaling, cat#2185, 1:500 dilution), anti-SNAI2 (Abcam, cat#ab27568, 1:200 dilution), and anti-p16 (CDKN2A, Abcam, cat#ab54210, 1:1000 dilution) antibodies were used.

**CRISPR-Cas9 editing with lentiviral constructs**. Cas9 gene encoding the *S. pyogenes* CRISPR associated protein 9 RNA-guided DNA endonuclease Cas9 was cloned under control of the human cytomegalovirus promoter into a lentiviral construct derived from pLenti6 (#V49610, Invitrogen) carrying a blasticidin resistance cassette. Upon packaging, the active virus was used to transduce the construct into P6 nHPEKs grown in CnT-Prime media (CELLnTEC) (Supplementary Fig. 4a). After 7 days of blasticidin at $0.625\,\mu g\,ml^{-1}$, blasticidin-resistant cells were further grown for 2 days in CnT-Prime media, analyzed for Cas9

expression, cryopreserved (into pellets of $2 \times 10^6$ cells) in liquid nitrogen for later use and assessed for editing using a sgRNA against PIG-A (Phosphatidylinositol Glycan Anchor Biosynthesis Class A) with the following sequence: 5′-TGGCGT GGAAGAGAGCATCA-3′. For editing assessment nHPEKs-Cas9 were infected with the PIG-A sgRNA and a control at a multiplicity of infection of 1. Cells were maintained with puromycin selection until day four and transduction efficiency was assessed by flow cytometry using the red fluorescent protein (RFP) reporter encoded on the lentiviral construct. If >90% RFP-positive cells were measured, expansion of cells continued without further puromycin selection. Gene-editing efficiency was assessed using the software TIDE (Tracking of Indels by DEcomposition)[71].

**CRISPR–Cas9 mini-pool screen with lentiviral constructs**. For the pooled sgRNA library, 2698 sgRNA sequences were selected for 540 genes. The library was constructed using chip-based oligonucleotide synthesis (Custom Array) to generate spacer-encoding fragments that were polymerase chain reaction (PCR)-amplified and cloned as a pool into the BbsI site of pNGx-LV-g003 lentiviral plasmid[72]. The sgRNA designs were based on published sequences[73] and five sgRNAs were selected per gene targeting the most proximal 5′ exons. Sequencing of the plasmid pool showed robust normalization with >90% clones present at a representation of ±fivefold from the median counts in the pool.

sgRNA libraries were packaged into lentiviral particles by growing HEK293T cells in T150 flasks (Corning, cat#3313,). For each flask, $2.1 \times 10^7$ cells were transfected 24 h after plating using 510.3 ml of TransIT reagent (Mirus, Madison, WI, cat#MIR2300,) diluted in 18.4 ml of Opti-MEM that was combined with 75.6 mg of the sgRNA libraries and 94.5 mg of lentiviral packaging mix (cat#CPCP-K2A, Cellecta, psPAX2 and pMD2 plasmids that encode Gag/Pol and VSV-G, respectively)[74]. Seventy-two hours post transfection, lentivirus was harvested, aliquoted, and frozen at −80 °C. Viral titer was measured by fluorescence-activated cell sorting in HCT116 cells and was typically in the range of $5 \times 10^6$ TU/ml.

For the screen, nHPEKs-Cas9 were expanded in CnT-Prime media to $3 \times 10^6$ cells in T300 flasks (TPP, Trasadingen, Switzerland) and transduced with the lentiviral sgRNA library (described in detail above) with a coverage of 5 sgRNAs/gene and a MOI of 0.5 aiming for coverage of on average 1000 cells/sgRNA. On day 0, each flask was infected with the lentivirus pool supplemented with $1\,\mu g\,ml^{-1}$ polybrene. After 24 h the culture media was replaced with fresh media containing $1\,\mu g\,ml^{-1}$ puromycin. Seventy-two hour after puromycin addition, cells were detached using Accutase and plated into new flasks stacks at $4.5 \times 10^6$ cells per stack. Transduction efficiency was assessed by flow cytometry and the experiment only continued if >95% cells were RFP positive. Cells were maintained in culture, split as needed to ensure confluence did not exceed 90% and at least $5 \times 10^6$ cell pellets were collected and kept in −80 °C after each split. Cell pellets were collected at days 12, 19, 28, 33, 38, 43, and 47 post infection.

Genomic DNA from cells was isolated using the PureLink Genomic DNA Mini Kit (Invitrogen, cat#K1820-02) and quantified using Picogreen (Invitrogen, cat#P11496) following manufacturer's instructions. Illumina sequencing libraries were generated using PCR amplification with primers specific to the genome integrated lentiviral vector backbone sequence. A total of eight 500 ng independent PCR reactions were performed per sgRNA transduced sample. PCR reactions were performed in a volume of 100 μl, containing a final concentration of 0.5 μM of each PCR primer (Integrated DNA Technologies, 6428 5′-AATGATACGGCGACC ACCGAGATCTACACTCG ATTTCTTGGCTTTATATATCTTGTG-3′ and INDEX 5′-CAAGCAGAAGAC GGCATACGAGATxxxxxxxxxxATTGTGGATG AATACTGCCATTTG-3′, where the Xs denote a 10 base PCR-sample specific barcode used for data demultiplexing following sequencing), 0.5 mM dNTPs (Clontech, cat#4030), 1× Titanium Taq DNA polymerase and buffer (Clontech, #639242). PCR cycling conditions were as follows: 1 × 98 °C for 5 min; 28 × 95 °C for 15 s, 65 °C for 15 s, 72 °C for 30 s; 1 × 72 °C for 5 min. The resulting Illumina libraries were purified using 1.8× SPRI AMPure XL beads (Beckman Coulter, #A63882) following the manufacturer's recommendations and qPCR quantified using primers specific to the Illumina sequences using standard methods. Illumina sequencing libraries were then pooled and sequenced with a HiSeq 2500 instrument (Illumina) with $1 \times 30$b reads, using a custom read 1 sequencing primer 5645 (5′-TCGATTTCTTGGCTTTATATAT CTTGTGGAAAGGACGAA ACACCG-3′), and a $1 \times 11$b index read, using the a custom indexing primer 6430 (5′-GATCTTGAGACAAATGGCAGTATTCATCCACAAT-3′), following the manufacturer's recommendations. A total of $1 \times 10^6$ reads were generated per transduced sample, resulting in an average of approximately a 500 reads per sgRNA.

Raw sequencing reads were aligned to the appropriate library using Bowtie[75] allowing for no mismatches and counts were generated. To assess effects on proliferation, the fold change of the unsorted cell population compared to the input library was generated. For gene-based hit calling, RSA[76] and average or maximal fold changes were calculated across all reagents for a given gene.

**CRISPR–Cas9 editing using electroporation**. Two gRNA per gene were designed on a DNA sequence flanking an exon and an intron as close as possible to TSS using the Design sgRNA for CRISPRko from the Broad Institute. A guide targeting an exon and the other an intron were chosen based on the Combined Rank

provided by the design tool with minimal predicted off-target binding activity. Guides were constructed by Integrated DNA Technologies (IDT, Coralville, IA). Supplementary Table 1 shows the sequences of all gRNA designed. A negative control gRNA was purchased from IDT (Alt-R® CRISPR-Cas9 Negative Control crRNA #1).

Recombinant Cas9 was produced in *Escherichia coli* containing two nuclear localization sequences and a His6 tag and purified with NiNTA chromatography, followed by size-exclusion chromatography. Two gRNAs per gene were initially combined with tracrRNA (IDT) at 2 µg ml$^{-1}$, heated at 95 °C for 5 min and left to cool down to room temperature (RT) before storing at −20 °C. Sample preparation for transfection was performed using the Neon® Transfection System 100 µL Neon® Tips (ThermoScientific, cat#MPK10025). For the transfection, gRNA/tracrRNA were mixed in a ratio 2:1 with rCas9 and Duplex buffer (IDT) and incubated 10 min at RT. Keratinocytes were collected and suspended in Buffer R (ThermoScientific) at $2.7 \times 10^7$ cells/ml and mixed with the gRNA/tracrRNA for 2–3 min with a final cell concentration of $2 \times 10^7$ cells/ml. Transfection was performed using the Neon transfection system (ThermoScientific, cat#MPK5000) and keratinocytes were immediately transferred to flasks containing CnT-Prime media (CELLnTEC) and incubated for 72 h.

**Construction of IRF2-HA induction systems**. IRF2 cDNA was obtained from the Invitrogen 96 Ultimate ORF Clone Lite Collection and was PCR amplified with using the following primer pair to add a 5′ CACC overhang and a C-terminal 3XHA tag:

CACCATGCCGGTGGAAAGGATGCGC
CTGCGGCCGCTTAAGCGTAATCTGGAACGTCATATGGATAGGATC
CTGCATAGTCCGGGACGTCATAGGGATAGCCCGCATAGTCAGGAACAT
CGTATGGGTAACAGCTCTTGACGCGGGCCTG

The CACC enabled directional TOPO cloning into pENTR/D-TOPO (ThermoFisher Scientific). Gateway LR (ThermoFisher Scientific) reaction was utilized to clone IRF2-3XHA into pINDUCER20[77].

The IRF2 gene was cloned under control of a doxycycline inducible promoter with a neomycin resistance cassette (pINDUCER20) and introduced, using lentiviral infection, in to aHPEKs previously knocked-down for IRF2 using the Neon® transfection system. aHPEKs were selected with G418 (6.25 µg ml$^{-1}$) for 7 days then treated with 1 µg ml$^{-1}$ Doxycycline for 3 days plus 2 days with 2 µg ml$^{-1}$ Doxycycline to induce the HA tagged-IRF2 overexpression.

**Quantification and statistical analysis**. The gene-set database was compiled from multiple sources including Reactome, NCBI Biosystems, and Gene Ontology. Enrichments were calculated using a hypergeometric overrepresentation test. Benjamini Hochberg–corrected P values were calculated for each gene set and combination of input gene list size. For RNA-seq data, GSA were performed using a python script that conduct a series of (non)parametric tests for each contrast. We performed all of the analyses using two gene rankings, where the first one is by fold-change only and the second is by the fold-change multiplied by the negative logarithm of the $p$ value; the latter version puts more weight on genes that move less, but significantly. Within a contrast, results were adjusted using the Benjamini–Hochberg method.

**RNA-sequencing**. Gene expression for keratinocytes was measured using RNA sequencing technology. Each experimental condition was performed in triplicate. RNA was isolated with Qiazol (Invitrogen, cat#79306), and purified with the miRNeasy kit (Qiagen, cat#217004). The amount of RNA was quantified with the Agilent RNA 6000 Nano Kit (Agilent Technologies, cat#5067-1511). RNA libraries were prepared using the Illumina TruSeq RNA Sample Preparation Kit v2 (Illumina, cat#RS-122-9001DOC) and sequenced using the Illumina HiSeq2500 platform following the manufacturer's protocol. Samples were sequenced in paired-end mode to a length of $2 \times 76$ bp. Images from the instrument were processed using the manufacturer's software to generate FASTQ sequence files. Read quality was assessed by running FastQC (version 0.10) on the FASTQ files. Sequencing reads showed excellent quality, with a mean Phred score higher than 30 for all base positions. Paired-end reads were mapped to the Homo Sapiens genome (GRCh38.p7) and the human gene transcripts from Ensembl v87 by using an in-house gene pipeline (https://www.ncbi.nlm.nih.gov /pubmed/27302131). Genome and transcript alignments were used to calculate gene counts based on Ensembl gene IDs. The raw RNA-sequencing counts are available in Supplementary Data 5 (LSCP versus HSCP) and Supplementary Data 6 (IRF2 KO versus control).

Gene counts were divided by the total number of mapped reads for each sample and multiplied by one million to obtain Counts Per Million (CPMs) to account for varying library sizes. Principal Component Analysis was used to detect possible outliers. Differential expression analysis was performed on the CPMs using a limma/voom workflow with R version 3.33 https://cran.r-project.org/doc/FAQ/R-FAQ.html. Genes with counts per million values below 1 for at least 60% of the replicates in both groups were excluded from the analysis. Differential genes were called using limma voom with the following contrasts: HSCP-HK versus LSCP-HK, LSCP-HK CT versus LSCP-HK IRF2 KD results are reported in terms of log$_2$ fold changes and negative log10 adjusted P values (Benjamini Hochberg FDR) in Supplementary Data 1.

**Chromatin immunoprecipitation of H3K27ac and sequencing**. Approximately, $4 \times 10^6$ cells were cross-linked with 10% formaldehyde at RT for 15 min. After quenching with 125 mM glycine for 6 min, cells were washed in ice-cold PBS (plus PI Cocktail tablet (Sigma) and 100 mM PMSF) and collected by scraping in Nuclei EZ lysis buffer (Sigma, plus PI and 100 mM PMSF). Nuclei were collected by centrifugation (500×$g$ for 5 min at 4 °C). In order to increase nuclei collection, pellets were further re-suspended in a Hypotonic Buffer (10 mM HEPES, 10 mM KCl, 0.1 mM EDTA, 0.1 mM EGTA, pH8.0) for 10 min in ice followed by NP-40 (0.5% v/v) and 20 s vortex at maximum speed. Nuclei were collected by centrifugation (14,000 rpm, 4 °C, 30 s), washed with Wash Buffer C (with PI) (truChIP Chromatin Shearing Kit, Covaris, cat#520154) and re-suspended in Shearing buffer D3 (plus PI and 100 mM PMSF) (truChIP Chromatin Shearing Kit, Covaris). Nuclei were vortexed for 3 rounds of 10 s and left 15 min in ice before freezing in −80 °C for later use. After thawing, 130 µl nuclei suspensions were transferred into microTUBE (Covaris) and sonicated using a Covaris E220 (10 min: duty cycle 2%, Intensity 5, Cycles per Burst 200). Samples were collected in 1.5 ml tubes, centrifuged at 10,000×$g$ for 10 min at 4 °C and supernatants transferred into 2 ml tubes. DNA concentrations were calculated after DNA isolation using Qiaquick PCR (Qiagen) using Qubit Nucleic Acid Quantification (Invitrogen). For ChIP a modified protocol of the Magna ChIP A/G (Merck, cat#17-10085) protocol was used. In brief 5 µg of mice anti-H3K27ac (Active Motif, cat#339685) or mAb IgG1 Isotype Control (Cell Signaling, cat#5415) were coupled with Protein A/G beads at 4 °C for 4 h. Coupled Ab-beads were mixed with 5 µg of samples and incubated overnight at 4 °C with rotation. Ab-beads were washed as per manufacturer's instructions and protein/DNA complexes eluted in Elution buffer (50 mM NaCl and 1% SDS in dH$_2$O) for 4 h at 65 °C. Reverse cross-linking was performed by adding RNAse for 30 min at 37 °C and 5 mM EDTA, 20 mM Tris pH7.5 and 100 µg ml$^{-1}$ Proteinase for another 1 h at 45 °C. Supernatants were separated and DNA extracted using the MiniElute PCR purification kit (Qiagen). Illumina sequencing libraries were prepared from the ChIP and Input DNAs by the standard consecutive enzymatic steps of end-polishing, dA-addition, and adaptor ligation. After a final PCR amplification step, the resulting DNA libraries were quantified and sequenced on Illumina's NextSeq 500 (75nt reads, single end). The raw ChIP-sequencing reads are available in the NCBI GEO database.

**Chromatin immunoprecipitation of BRD4 and sequencing**. ChIP-Seq was performed by Active Motif Inc. (Carlsbad, CA, USA) as follows. Frozen cells ($9 \times 10^6$) were thawed and fixed with 1% formaldehyde for 15 min and quenched with 0.125 M glycine. Chromatin was isolated by the addition of lysis buffer, followed by disruption with a Dounce homogenizer. Lysates were sonicated and the DNA sheared to an average length of 300-500 bp. Genomic DNA (Input) was prepared by treating aliquots of chromatin with RNase, proteinase K and heat for de-crosslinking, followed by ethanol precipitation. Pellets were resuspended and the resulting DNA was quantified on a NanoDrop spectrophotometer. Extrapolation to the original chromatin volume allowed quantitation of the total chromatin yield. An aliquot of chromatin (30 µg) was precleared with protein A agarose beads (Invitrogen). Genomic DNA regions of interest were isolated using 4 µg of antibody against BRD4 (Bethyl, cat# A301-985A100, Lot A301-985A100-6). Complexes were washed, eluted from the beads with SDS buffer, and subjected to RNase and proteinase K treatment. Crosslinks were reversed by incubation overnight at 65 °C, and ChIP DNA was purified by phenol-chloroform extraction and ethanol precipitation. Quantitative PCR (QPCR) reactions were carried out in triplicate on specific genomic regions using SYBR Green Supermix (Bio-Rad). The resulting signals were normalized for primer efficiency by carrying out qPCR for each primer pair using Input DNA. Illumina sequencing libraries were prepared from the ChIP and Input DNAs by the standard consecutive enzymatic steps of end-polishing, dA-addition, and adaptor ligation. After a final PCR amplification step, the resulting DNA libraries were quantified and sequenced on Illumina's NextSeq 500 (75 nt reads, single end). The raw ChIP-sequencing reads are available in the NCBI GEO database.

**Chromatin immunoprecipitation of IRF2-HA and sequencing**. Chromatin immunoprecipitations were performed using the ChIPmentation protocol as described[78] with minor changes. The cells (no doxycycline IRF2 KD + 3xHA-IRF2 aHPEKs and doxycycline-treated IRF2 KD + 3×HA-IRF2 aHPEKs) were trypsinized and 1 million cells were used for each replicate. Cells were fixed by adding formaldehyde to a final concentration of 0.8% in a crosslinking buffer (0.1 M NaCl, 1 mM EDTA, 0.5 mM EGTA, 50 mM HEPES, pH 8.0) and incubated for 5 min on a rocker at 4 °C. The fixation was terminated by incubating with 2.5 M Glycine for 5 min on a rocker at RT for 5 min. The cells were washed in ice-cold PBS and resuspended in hypotonic buffer (20 mM Tris-HCl, 10 mM NaCl, 3 mM MgCl$_2$) supplemented with PIs and 0.5 mM PMSF and incubated on ice for 10 min. NP-40 (final 0.5%) was added to the mixture and vortexed for 20 s. The nuclei were pelleted by centrifugation at 14000 rpm for 30 s at 4 °C. The pellet was resuspended in 1 ml of 1× Wash Buffer C (truChIP Chromatin Shearing Kit) and incubated at 4 °C for 10 min in a rocker. The nuclei were pelleted by centrifugation at 1700×$g$ for 5 min at 4 °C and subsequently resuspended in shearing buffer D3 (truChIP Chromatin Shearing Kit) with PIs and 0.5 mM PMSF, vortexed thrice and incubated at 4 °C for 15 min in a rocker. Nuclear pellets were collected by centrifugation at 2000xg for 5 mins at 4 °C and resuspended in 130 µl of TE (10 mM Tris and

1 mM EDTA) with 0.1% SDS and DNA was sheared at 4 °C using a covaris LE220 sonicator (Covaris, Woburn, MA) for 4 min with 200 cycles per burst, 15% duty factor and a peak incident power of 300. Sonicated lysates were supplemented with salts and detergents to a final concentration of 1% Triton X-100, 150 mM NaCl and 0.1% Na-deoxycholate. The chromatin was then cleared by centrifugation at 10000xg for 10 min and incubated with 5 μl of Dynabeads™ Protein A (Thermo Fisher Scientific, Waltham, MA) for one hour to preclear the lysate. After collecting the input, the supernatant was incubated with end-over-end rotation overnight at 4 °C with Dynabeads™ Protein A magnetic beads prebound with HA antibody (Cell Signaling, cat#3724, 0.2 μg). The HA antibody was bound to the beads by incubating the beads with the antibody in 200 μl of bead binding buffer (TE, 0.2% Igepal) in a rotor at 4 °C for 2 hours. Beads were washed once with low salt wash buffer (250 mM NaCl, 10 mM Tris-HCl, 1 mM EDTA, 0.1% SDS, 0.1% Na-deoxycholate, 0.1% Triton X-100), once with wash buffer containing 500 mM NaCl, once with LiCl wash buffer (20 mM Tris pH8.0, 1 mM EDTA, 250 mM LiCl, 0.5% NP-40, 0.5% Na-deoxycholate), once with TE, 0.1% Triton X-100 and twice with ice cold 10 mM Tris-HCl. The beads and the input were then incubated at 37 °C for 12 min with the Tagment DNA enzyme (Illumina, San Diego, CA), following tagmentation, the beads were washed twice with the low salt wash buffer and DNA was eluted TE Buffer, 250 mM NaCl and 0.3% SDS. Cross-links were reversed by incubation first at 55 °C and continued with protein digestion with the addition of Proteinase K for 10 h at 64 °C. DNA was purified using DNA Clean & Concentrator-5 (Zymo Research, Irvine, CA).

Library preparations were performed as described with minor changes[79]. The amplified libraries were cleaned, and size selected using AMPure XP (Beckman Coulter, Indianapolis, IN). The libraries were sequenced using NextSeq® 500/550 High Output Kit v2 (75 cycles) (Illumina) in a NextSeq 550 (Illumina).

**ATAC-sequencing.** ATAC-seq (Assay for Transposase-Accessible Chromatin) was performed using $3 \times 10^5$ HPEKs that were seeded in 6-well plates, washed once with 500 μl of PBS (on plate) and detached using Accutase (CELLnTEC). Cells were centrifuged at 500×g and cell pellets placed on ice. Cells were resuspended in 1 ml lysis buffer (10 mM Tris-HCl pH7.5, 10 mM NaCl, 3 mM MgCl₂, 0.1% IGEPAL CA-630, 1 mM PMSF and PI cocktail). Cell lysates were split into 2 × 1.5 ml tubes (for an equivalent of $1.5 \times 10^5$ HPEKs per tube). The suspension of nuclei was then centrifuged for 10 min at 500×g at 4 °C, followed by the addition of 50 μl transposition reaction mix (25 μl TD buffer, 2.5 μl Tn5 transposase and 22.5 μl nuclease-free H₂O) of Nextera DNA library Preparation Kit (Illumina, 96 samples, cat#FC-121-1031). Samples were then incubated for tagmentation at 37 °C for 30 min. The reaction was stopped using 50 μl Stop Buffer (30 mM EDTA, pH8.0 and 90 mM NaCl)[37,79]. DNA was isolated using a MinElute Kit (QIAGEN). ATAC-seq libraries were subjected to 15 cycles of amplification. The Agencourt AMPure XP PCR purification system (Beckman Coulter) was used for purification of PCR amplicons. Library concentration was measured using Qubit dsDNA HS Assay Kit (Invitrogen, cat#Q32854) according to the manufacturer's instructions. Library integrity was checked by gel electrophoresis. Fragment size distribution was checked using High-sensitivity DNA chip (Agilent, cat#5067-4626) for 2100 Bioanalyzer. Finally, the ATAC library was sequenced on an Illumina HiSeq2500 using HiSeq v4 Sequencing Kit (2 × 76 cycles) (cat#FC-401-4002) according to the manufacturer's instructions.

**ChIP-sequencing data analysis.** All coordinates and gene annotations in this study were based on human reference genome assembly HG19, GRCh37 (ncbi.nlm.nih.gov/assembly/2758/) and RefSeq genes. The genome and transcriptome gene gtf were obtained from (ftp://igenome:G3nom3s4u@ussd-ftp.illumina.com /Homo_sapiens/ UCSC/hg19/Homo_sapiens_UCSC_hg19.tar.gz). The HG19 NCBI RefSeq gene table was downloaded directly from the UCSC genome browser from their table browser utility (http://genome.ucsc.edu/cgi-bin/hgTables). The HG19 ENCODE blacklist was used to filter genomic regions and was obtained from (https://sites.google.com/site/anshulkundaje/ projects/blacklists).

All coordinates and gene annotations in this study were based on human reference genome assembly HG19, GRCh37 (ncbi.nlm.nih.gov/assembly/2758/) and RefSeq genes. The genome and transcriptome gene gtf were obtained from (ftp://igenome:G3nom3s4u@ussd-ftp.illumina.com /Homo_sapiens/ UCSC/hg19/ Homo_sapiens_UCSC_hg19.tar.gz). The HG19 NCBI RefSeq gene table was downloaded directly from the UCSC genome browser from their table browser utility (http://genome.ucsc.edu/cgi-bin/hgTables). The HG19 ENCODE blacklist was used to filter genomic regions and was obtained from (https://sites.google.com/site/anshulkundaje/ projects/blacklists).

All paired-end ChIP-seq datasets were aligned using Bowtie2 (version 2.2.8) to build version NCBI37/HG19. Alignments were performed using all default parameters. We used the MACS version 1.4.2 (Model based analysis of ChIP-seq)[80] peak finding algorithm to identify regions of ChIP-seq enrichment over background. A p value threshold of enrichment of 1e−9 was used for all datasets.

IRF2 ChIP-seq data were aligned using Bowtie2 (version 2.3.4.1) again with default parameters.

**ATAC-seq data processing.** All paired-end ATAC-seq datasets were aligned using Bowtie (version 2.2.8) to build version NCBI37/HG19 with the following

parameters: --end-to-end --sensitive --no-unal --no-discordant --mm --met-stderr. To process aligned ATAC-seq data, we used the RIESLING pipeline[81] developed jointly with the Gordon laboratory. Briefly, aligned bams were filtered for low quality reads, duplicate reads were removed, reads were sorted using samtools, mitochondrial reads were removed, and further filtering against the ENCODE blacklist. Peaks were called using MACS1.4.2 which does not take into account paired end fragment information. A threshold of p value < 1e−9 was used to call enriched regions.

**RNA-seq data processing.** For integrated chromatin analysis, RNA-seq data were reprocessed to align to and quantify the HG19 RefSeq transcriptome. Fastq files were aligned to the HG19 genome using HiSat2 (version 2.0.4) with default parameters. Transcripts were quantified and FPKM values were generated using cuffquant and cuffnorm from the cufflinks suite (version 2.2.1) (http://cole-trapnell-lab.github.io/cufflinks/) on the HG19 RefSeq transcriptome. MetaCore Interactome enrichment tool was used (https://clarivate.com/products/metacore/) to predict TF connectivity analysis.

**Calculating read density.** We calculated the normalized read density of a ChIP-seq or ATAC-seq dataset in any genomic region using the Bamliquidator (version 1.0) read density calculator (https://github.com/BradnerLab/pipeline/wiki/bamliquidator). Briefly, reads aligning to the region were extended to 200 bp and the density of reads per base pair (bp) was calculated. The density of reads in each region was normalized to the total number of million mapped reads producing read density in units of reads per million mapped reads per bp (rpm/bp). The AUC of read occupancy in each region was reported simply as the number of reads divided by the total number of million mapped reads producing a AUC measurement in total reads per million (rpm).

**Mapping of active chromatin landscapes.** We used H3K27ac ChIP-seq to define the active chromatin cis-regulatory landscape across human keratinocytes. Briefly, we took the union of all H3K27ac enriched regions in any single H3K27ac dataset in either HSCP-HK or LSCP-HK samples (n of 3 each). This yielded 24,614 discreet H3K27ac regions in the genome. To compare the similarity of H3K27ac at these regions between HSCP-HK and LSCP-HK samples, at each region on a per sample basis, we first quantified H3K27ac AUC (rpm). We next normalized on a per sample basis H3K27ac signal to the median region AUC. Similiarity between samples was calculated using a Pearson correlation and samples were clustered based of a correlation distance matrix (1 − correlation) (Supplementary Fig. 2).

HG19 RefSeq genes were considered active if they possessed an H3K27ac enriched region within the ±1 kb of their TSS in at least one H3K27ac dataset and their expression was greater than or equal to 10 FPKM in at least one sample. Using these criteria, we defined 7278 active HG19 RefSeq transcripts. As genes often have more than one TSS, these 7278 active transcripts corresponded to 5846 common gene names.

We sought to examine relationships between H3K27ac, BRD4, and mRNA levels at active genes between HSCP-HK and LSCP-HK. First, we quantified H3K27ac and BRD4 AUC occupancy at all active genes. For each gene, we assigned H3K27ac and BRD4 occupancy within the ±50 kb window flanking the TSS. All active genes were ordered based on their $log_2$ fold change in BRD4 AUC between HSCP-HK and LSCP-HK. Using bins of 100 genes, the average $log_2$ fold change in H3K27ac AUC or mRNA levels was plotted (Fig. 2a). Error bars represent the 95% confidence interval of the mean determined by empirical resampling with replacement (1000 permutations).

To compactly display ChIP-seq and ATAC-seq signal at individual genomic loci, we used a simple meta representation[16]. For all samples within a group, ChIP-seq or ATAC-seq signal is smoothed using a simple spline function and plotted as a translucent shape in units of rpm per bp. Darker regions indicate regions with signal in more samples. An opaque line is plotted and gives the average signal across all samples in a group.

**Core transcriptional regulatory circuitry analysis.** We first identified TFs in keratinocytes that were actively expressed and regulated by a proximal H3K27ac defined active cis-regulatory element (n = 126). We next filtered these TFs for protein–protein interactions as defined by the STRING protein–protein interaction database[42]. Protein–protein interactions were defined in STRING as having a medium confidence >0.400 interaction defined by all active interaction sources. Markov Chain Linkage with an inflation parameters was then used to cluster TFs by protein–protein interaction. These resulted in 60 remaining TFs across 6 clusters (Fig. 3a).

To define the transcriptional circuitry of keratinocytes and regulatory connectivity of keratinocyte KTFs, we developed the CRC software package, an updated version of the COLTRON CRC software initially reported in ref. [16]. Using the 60 TFs previously defined, we defined a regulatory interaction as a TF binding to a nucleosome free region inside an active cis-regulatory element. Cis-regulatory elements were previously defined by H3K27ac and nucleosome free regions are defined as ATAC-seq peaks contained within H3K27ac defined cis-regulatory regions. Within these regions, the CRC software uses FIMO[43] to find enriched (q value < 1e−5) TF motif occurrences. CRC defines TF motifs using a custom

database aggregating published TF motif position weight matrices from TRANSFAC[40], JASPAR[39], and[41]. Thus for each TF, a set of predicted binding sites at ATAC-seq peaks within H3K27ac regions is provided. To assess the similarity of TF motif occupancy profiles, motifs for all 60 TFs were extended by 50 bp and a binary occupancy matrix of all discreet regions in the genome with at least 1 predicted motif was assembled ($n = 49,986$ regions). For each TF the pairwise pearson correlation was calculated and all TFs were hierarchically clustered (Supplementary Fig. 3e).

For each TF, we defined its inward and outward regulatory connectivity (Supplementary Fig. 3a). The inward regulatory connectivity is defined by the number of unique TFs that are predicted to bind to its proximal *cis*-regulatory elements. Likewise, for each TF, the outward regulatory connectivity is defined by its binding to the *cis*-regulatory elements of all other TFs. A TF binding to its own *cis*-regulatory element is considered both an inward and outward regulatory connection. Thus for each of the 60 TFs, a maximum of 60 inward and 60 outward connections are possible. The inward and outward regulatory connectivity for all 126 active keratinocyte TFs are shown with those 60 in clustered protein–protein networks highlighted (Supplementary Fig. 3b). The total (inward + outward) connectivity for TFs in clusters 1 and 2 was compared to those TFs not in protein–protein interaction networks as expressed as a fraction of maximum possible connectivity (Supplementary Fig. 3c). The statistical significance of the difference in distributions was assessed using a one-sided Wilcoxson Rank-Sum test (**$p$ value < 1e−5). Within either cluster 1 or cluster 2, the total maximum possible connectivity for TFs within the cluster and for TF–TF interactions with other actively expressed TFs was assessed (Supplementary Fig. 3d). The statistical significance of the difference in distributions was assessed using a one-sided Wilcoxson Rank-Sum test (*$p$ value < 1e−3).

Similar to regulatory inward and outward connectivity, we defined for each TF the BRD4 IN and OUT degree (Fig. 3c). For each TF, BRD4 IN degree is quantified as the amount of BRD4 at its proximal *cis*-regulatory regions. For each TF, BRD4 OUT degree is quantified as the amount of BRD4 at its predicted binding sites across all other *cis*-regulatory regions. For each TF, we took the log$_2$ fold change in BRD4 OUT degree between HSCP-HK and LSCP-HK. We ranked all 60 TFs by log$_2$ change in BRD4 OUT degree (Fig. 3d). Error bars represent the standard error of the mean.

**IRF2-binding analysis**. ChIP-seq was performed in duplicate for the HA antibody in either HSCP-HKs engineered with doxycycline inducible IRF2-HA or parental unengineered HSCP-HKs. HA peaks were called versus whole-cell extract background using MACS1.4.2. High confidence IRF2 peaks were determined as those HA peaks found in both IRF2-HA expressing cells and absent in the parental unengineered cells. This resulted in 4135 peaks of which 2068 overlapped an ATAC-seq peak in at least one of the samples.

For both IRF2 peaks with and without overlapping ATAC-seq, we performed de novo motif finding using the meme-chip software (version 4.11.4) with the following parameters (-meme-nmotifs 5 -spamo-skip -db VertebratePWMs.txt). For the -db argument, the same table of vertebrate TF position weight matrices used in the transcriptional regulatory circuitry analysis was supplied (https://github.com/linlabcode/CRC/blob/master/crc/annotation/ VertebratePWMs.txt). For both sets of regions, the top identified motif is reported.

BRD4 out degree was calculated as before, but only on edges overlapping a high confidence IRF2-binding site. The average change in BRD4 out degree for these IRF2 ChIP-validated edges is reported and compared to IRF2 predicted edges and all edges in the *cis*-regulatory landscape. Error bars represent the standard error of the mean (Fig. 6d).

We used an approach previously developed[82] to quantify IRF2 signal proximal to each gene. Briefly, background subtracted IRF2 occupancy (AUC) was calculated at all 2068 high confidence regions. Genes were ranked by their combined promoter (±1 kb TSS) and distal IRF2 occupancy. Distal IRF2 occupancy represents the AUC of IRF2 peaks within ±50 kb of a gene's TSS that are not overlapping another genes' TSS. IRF2 signal was detected at 855 unique genes. These were ranked by cumulative promoter + distal IRF2 AUC (Fig. 6f). Gene-set enrichment analysis was used on these 855 genes ranked by IRF2 AUC to identify signatures with leading edge enrichment (pulled from the mSigDB C2 set). Log$_2$ fold expression changes upon IRF2 KD or between HSCP-HKs and LSCP-HKs were calculated at these 100 genes and for all other active genes without any evidence of IRF2 binding. The statistical significance of the difference between distributions was tested using a two-tailed *t* test (Fig. 6e). Finally, the top 100 IRF2 targets were queried using the mSigDB investigate gene sets tool against the C2 curated gene sets. The significance of enrichment for the top ten gene sets are shown in Fig. 6g.

**Reporting summary**. Further information on research design is available in the Nature Research Reporting Summary linked to this article.

## Data availability

High throughput sequencing data that support the findings of this study have been deposited in NCBI GEO under the overall accession code: GSE135680 (https://www.ncbi.nlm.nih.gov/geo/query/acc.cgi?acc=GSE135680) ATAC-seq in HSCP-HKs and LSCP-HKs can be found under: GSE135675. H3K27ac and BRD4 ChIP-seq in HSCP-HKs and LSCP-HKs can be found under: GSE135676. IRF2-HA ChIPmentation can be found under: GSE135677. RNA-seq of baseline HSCP-HKs and LSCP-HKs can be found under: GSE135679. RNA-seq expression level count data for IRF2-KD experiments are available in Supplementary Data Files 5 and 6. Read depth for all samples is provided in Supplementary Data 7.

## Code availability

All manuscript specific custom analysis scripts and code can be found at: https://github.com/charlesylin/keratinocyte_scripts. The transcriptional core regulatory circuitry analysis code can be found at https://github.com/linlabcode/CRC.

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

## Acknowledgements

We thank Philipp Hoppe, Lapo Morelli. and Daniela Stauffer for help with CRISPR guide design and editing methodology. Walter Carbone for high throughput sequencing and

informatics and Alicia Lindeman for high throughput sequencing of CRISPR libraries. Nicole Renaud and Florian Kiefer for support in developing the chromatin analysis workflow. Kayhangabriel Akyel and Oleg Iartchouk for enabling the BRD4 ChIP analysis at Active Motif. William Tschantz for supply of Cas9 protein. Svenja Ley for histology and staining on the 3D human skin model.

## Author contributions

F.L., N.M., C.Y.L. and S.K. conceptualized the study, designed, and analyzed the experiments: J.R-H. and J.A. generated the reagents for the pooled CRISPR screen: N.M., D.E., D.H., C.R. and C.K. designed and executed the CRISPR–Cas9 mini-pool screen: N.M. and R.T. designed and executed the H3K27ac ChIP: F.L. and N.M. performed ATACSeq: J.K. and G.R. performed RNASeq: H.R. and A.A. designed and executed the 3D human skin model experiments: G.S. performed the keratinocyte migration assays. N.M., G.S. and A.S. performed all other keratinocyte experiments: C.K. performed integrative analysis across the data sets: S.B. analyzed BRD4 ChIP experiments: C.G.K., F.N. and G.R supervised all RNASeq, ChIPSeq, and ATACSeq analysis: C.Y.L. and T.C.B. performed the core transcriptional circuitry analysis: C.M.O. and H.E. Sheppard built the IRF2 construct: N.M. generated the samples and S.J. performed the IRF2 ChIP. S.E. analysed the IRF2 ChIP: M.F. established the collaboration S.K., F.L., N.M., C.Y.L., C.K. and T.B. wrote and revised the paper.

## Competing interests

Most authors (with the exception of C.Y.L., T.B., S.J., C.M.O., H.E.S., S.E., all Baylor, TX) were employees of Novartis during the course of this work, as indicated in the affiliations. C.Y.L. received sponsored travel from Novartis and is a shareholder and inventor of IP licensed to Syros Pharmaceuticals. Some authors own shares in Novartis. The remaining authors declare no competing interests.
