## [Peer Review File · Nature Communications]

Reviewers' comments:

Reviewer #1 (Remarks to the Author):

Here the authors investigated the role of specific transcription factors in reshaping the keratinocyte active chromatin landscape to control proliferation potential of somatic stem cells. The authors, using RNA seq, epigenomic profiling (ATAC, H3K27ac, etc.), computational circuitry analysis and genetic validation, aim to identify complex network involving transcription factors controlling stemness. They compared human keratinocytes with high stem cell potential (low passage) and keratinocytes with limited stem cell function (high passage). Overlapping the data, they identified putative transcription factors acting possible as regulators of somatic stem cell proliferative potential. Using CRISPR-mediated genetic loss of function they confirmed that part of these transcription factors, among them IRF2, are implicated in controlling stemness of somatic stem cells. They, using in vitro experiments and 3D skin culture, demonstrated that IRF2 loss sustained somatic stem cell function in normal human keratinocytes. The article is interesting, the role of IRF2 in controlling keratinocytes proliferating potential is novel and interesting, yet the link between IRF2 and stemness should be further investigated. The authors performed many experiments with adequate controls and statistical analysis, yet they need to further confirm their findings at functional level to justify their claims and better characterizing the experimental model they are using.

Major point: The model used in the study, that is serial passaging of primary keratinocytes, have been already described by many articles as a strategy to induce replicative senescence in vitro in primary cells, this process is not keratinocytes specific but for every primary cell lines (the well known Hayflick limit). The major point that the authors need to clarify is the relation between "human keratinocytes with high stem cell potential"- HSCP-HK and/or "human keratinocytes with low stem cell potential" - LSCP-HK and the undergoing senescence they are inducing. In the rescue experiments, are they targeting the somatic stem cells? Indeed, IRF2 could regulates transient amplifying cells, known to be committed to differentiation, and not the progenitors/stem cells.

In line with this observation, I think that the authors should either provide additional experiments showing that they are indeed targeting somatic stem cells/progenitor cells or modify the title and the text including the concept that they are comparing proliferating versus senescent cell phenotype.

Additional points:

- In Fig 1 the authors should better characterize the HSCP and LSCP their are using. For example, they should evaluate holoclones/meroclones and paraclones between the two populations. Since the holoclone are generated only by somatic stem cells/precursors, it will be crucial to see that the LSCP are indeed able to produce holoclones. This is also important in the experiment shown in Fig 5C (also 5A and B).

- Cellular senescence, beside being associated to an increase of beta-galactosidase staining, is also associated to other markers: expression of p16, PML, hypo-phosphorylation of Rb, decrease of telomerase activity, accumulation of DNA damage. How the LSCP cells express this markers in comparison with HSCP? Does this markers are altered in relation to IRF2 expression? Specifically, LSCP keratinocytes, after IRF2KO, have reduced senescence markers? Do they form holoclones?

- The reduced migration capacity observed in the LSCP cells is a direct effect of IRF2 modulation? Could be p63 down-regulation been responsible for this? Is it reported by many articles that p63 controls a set of genes related to cell adhesion and cell-cell contact.

- The majority of the 3D skin images shown are low quality (ie. 5B, out of focus and/or damaged). In general, measurement of the 3D skin thickness is not sufficient to claim an increase in stem cell potential. The authors should also evaluate the "quality" of the 3D reconstructed skin. For instance, they should evaluate basal layer/upper layers markers, Ki67+ cells, p63+ cells, evaluate if the switch between proliferating/differentiating layer (K14/K10) occurs normally in the different conditions shown

(both for Fig 5 and Fig 7), etc.

- Fig 7 F is not clear at all. Based on my experience this experiment needs to be re-done. Why did they include the dotted line? The authors should add K14 staining as a basal layer marker. Why does dark staining appear in the dermis? Is the antibody specific? Why do LSCPs do not form the cornified layer (also in Fig 7E)? Again, without specific markers for each layer it is not possible to evaluate what is shown in the 3D skin culture in all the experiments presented.

- It has been demonstrated that IRF2KO gives a psoriatic-like phenotype in mouse skin. Is the phenotype observed in IRF2KO keratinocytes due to production of cytokines?

Reviewer #2 (Remarks to the Author):

This is a well-performed study to identify core regulatory circuitry (CRC) that dictates cell identity in primary keratinocytes with high stem potential (HSCP-HK) and those with low stem cell potential (LSCP-HK). The unbiased approach used to identify CRC TFs is novel and rigorous, and the findings are novel with potentially high impact in the field of epidermal biology. Comments/questions that I have for the authors are mostly minor and are listed below:

1. The model used to establish LSCP-HKs, using serial passaging, appears to recreate the phenotype/function of LSCP-HKs, but the distinction between LSCP-HKs and simply senescent cells, if in fact there is a distinction, should be better explained in the text. Namely, in normal human epidermis, is the process of differentiation, stratification, and eventual death of keratinocytes identical to or at least highly overlapping with the process of cellular senescence? How confident are we that the serial passaging model is truly mimicking the *in vivo* process of keratinocyte differentiation as these cells rise above the basal layer? Note to authors that these concerns are largely assuaged by the transcriptome/epigenome/CRISPR data presented, and by the reversibility of LSCP-HK function with IRF2 KO, so simply a better explanation of how differentiation relates to senescence would be appreciated to those who are not epidermal biologists.

2. Fig. 3A: What cells are being analyzed? This information does not appear to be stated anywhere in the text, methods, figure, or figure legend. Was the bioinformatic analysis performed on HSCP-HKs only?

3. Page 11, line 9: How do the authors justify focusing most of their studies on IRF2 as the master regulator? For example, BRD2 appears also to be a master regulator based on the data. Is this because BRD2 is not a TF, or because it does not alter migratory properties?

4. Fig. 5D, 7E and 7F: Some increased descriptive input from your pathologist might help the text on p12 (second paragraph). The authors describe differences in the epidermal component based on rather basic features such as "thickness", "cellularity", and "organization". What differences are present in the various layers? Have the authors scored the thickness of the stratum corneum in the various conditions? One might expect with greater differentiation that the corneal layer would be thinner perhaps in the IRF2 knockouts, though it does not appear to be. Some discussion on this point might help. Additionally, the stratum(s) between the basal and corneal layers appear less differentiated (more "cellular" on the words of the authors) in the IRF2 knockouts than CT counterparts. One might express this in terms of nuclear/cytoplasmic ratio, or nuclei/unit area, and likely this could be quantified. An attempt at illustrating differentiation with Keratin 10 is made in Fig. 7F, however it is not easy to see with the H&E counterstain. This would be better visualized with a minimal (e.g. hematoxylin-only) counterstain.

5. Page 14, paragraph 2: It is hard to interpret what the authors are saying. They state that IRF2KO in mice has a psoriatic phenotype, however it appears that they are saying that IRF2 KO in this model system reverts LSCP-HK cells to a less psoriatic phenotype. This seems contradictory and some clarification here is needed.

6. Page 18, line 6: It is interesting that IRF2 transcript levels do not differ between LSCP-HK and HSCP-HK cells given data that suggests an important anti-stemness role. If a good antibody to IRF2 exists, IHC or IF would be able to show whether IRF2 protein levels change as keratinocytes differentiate from basal to upper layers in their epidermal-dermal 3D model.

Reviewer #3 (Remarks to the Author):

The manuscript "IRF2 is a master regulator of human keratinocyte stem cell fate" by Mercado et al. identifies increased IRF2 expression as an antagonist of stem cell function in human keratinocytes. The manuscript is well written with appropriate experiments performed and presented. A few minor concerns that could be addressed to strengthen the manuscript are listed below:

1. Since a comparison between human and mice IRF2 is made, it would be informative to state what the % conservation of IRF2 is, and what the conservation means (or does not mean)
2. "Some of the top predicted TFs did not confirm in the screen most likely due to technical limitations with CRISPR editing of primary cells, current chromatin profiling techniques and assumptions required to build the core transcriptional algorithm."—This statement is unclear and confusing and requires some elaborate explanation regarding the indicated limitations.
3. It would be interesting to see the effects of depleting both YY1 and SNAI2 simultaneously in the experimental systems described.
4. It will be appropriate to provide the wound healing assay images along with quantification in Fig 1C.
5. In Fig 1C Y-axis - 'relative wound density' term is confusing. Should it be wound healing capacity or % migration?

Author's replies (in red) to Reviewers' comments (black):

Reviewer #1 (Remarks to the Author):

Here the authors investigated the role of specific transcription factors in reshaping the keratinocyte active chromatin landscape to control proliferation potential of somatic stem cells. The authors, using RNA seq, epigenomic profiling (ATAC, H3K27ac, etc.), computational circuitry analysis and genetic validation, aim to identify complex network involving transcription factors controlling stemness. They compared human keratinocytes with high stem cell potential (low passage) and keratinocytes with limited stem cell function (high passage). Overlapping the data, they identified putative transcription factors acting possible as regulators of somatic stem cell proliferative potential. Using CRISPR-mediated genetic loss of function they confirmed that part of these transcription factors, among them IRF2, are implicated in controlling stemness of somatic stem cells. They, using in vitro experiments and 3D skin culture, demonstrated that IRF2 loss sustained somatic stem cell function in normal human keratinocytes. The article is interesting, the role of IRF2 in controlling keratinocytes proliferating potential is novel and interesting, yet the link between IRF2 and stemness should be further investigated. The authors performed many experiments with adequate controls and statistical analysis, yet they need to further confirm their findings at functional level to justify their claims and better characterizing the experimental model they are using.

Author's reply to general remarks of reviewer 1

As requested we have performed an additional experiment to strengthen mechanistic understanding of how IRF2 is influencing keratinocyte cell state. In short, IRF2 has been removed by CRISPR-Cas9 editing then replaced with a tagged-IRF2 protein. This enabled IRF2 ChIP so that we now experimentally identify IRF2 binding sites instead of sole reliance on predicted binding through motif analysis. We find for the first time in human keratinocytes that IRF2 binds to and regulates gene pathways associated with interferon response and antigen presentation including beta macroglobulin, the immunoproteasome, and HLA-C. Loss of IRF2 leads to a downregulation of these genes followed by a selective downregulation of keratinization (differentiation) genes and an upregulation of cell cycle gene expression. Although the mechanistic link between keratinocyte stemness and IRF2 regulation of interferon response and antigen presentation will require further investigation, this study now validates the IRF2 prediction in transcriptional circuitry (Fig. 6d), identifies IRF2 direct targets, and characterizes the functional response to IRF2 perturbation.

Major point: The model used in the study, that is serial passaging of primary keratinocytes, have been already described by many articles as a strategy to induce replicative senescence in vitro in primary cells, this process is not keratinocytes specific but for every primary cell lines (the well-known Hayflick limit).

The major point that the authors need to clarify is the relation between "human keratinocytes with high stem cell potential"-HSCP-HK and/or "human keratinocytes with low stem cell potential" -LSCP-HK and the undergoing senescence they are inducing. In the rescue experiments, are they targeting the somatic stem cells? Indeed, IRF2 could regulates transient amplifying cells, known to be committed to differentiation, and not the progenitors/stem cells.

In line with this observation, I think that the authors should either provide additional experiments showing that they are indeed targeting somatic stem cells/progenitor cells or modify the title and the text including the concept that they are comparing proliferating versus senescent cell phenotype.

Model System

Our aim was to capture new regulators of keratinocyte stem cell function. We considered that an unbiased approach would enable such discovery and our ambition was to perform genome-wide transcriptional and epigenomic profiling of primary human keratinocytes. Such an analysis required $> 10^8$ primary keratinocytes with high stem cell potential and the same with low stem cell potential. The question of how we would experimentally obtain sufficient numbers in the most physiologically relevant fashion was the subject of much debate and pilot experiments as follows:-

- FACs analysis for surface stem cell markers was investigated in pilot experiments but has the downside that we already bias the analysis by using previously described stem cell markers which may not reflect stem cell function. Most importantly we found keratinocytes sorted on surface expression of such markers did not show greater clonogenicity and as we put our focus on stem cell function this method was not pursued further.
- Differentiation induced with calcium was another option but was not pursued as we thought that it would prove difficult to distinguish calcium induced signaling from loss of stem cell function. In addition, calcium rapidly drives terminal differentiation with significant changes in nuclear morphology which may have influenced our ability to obtain equivalent epigenomic analyses across the populations.
- Serial passaging was selected for the study for several reasons. We accept that replicative senescence is seen in many primary cell types and a phenomenon not restricted to keratinocytes. However in primary keratinocytes, a

close association exists between replicative senescence and differentiation which has been described in several studies. Such an association is also seen in vivo, where in aged mice an increase of senescence marker (p16) accompanies an increase in the differentiation marker (involucrin). Also, we did not passage the keratinocytes to a point where they were no longer dividing but even the low stem cell potential cells were still able to generate some colonies in the clonogenic assay, migrate and bind to matrix coated dishes (Figure 1).

We agree that we did not explain this aspect of the model sufficiently in our first submission and have rewritten this section of the introduction with additional references to explain the choice of the model and the known link between replicative senescence and differentiation (Highlighted text at the bottom of page 4, top of page 5). In addition we have added images of high and low stem cell potential cells in Figure 1C so that the reader can see the morphology of the cells and particularly that the low stem cell potential cells retained migratory function when sampled for the profiling experiments

HSCP vs LSCP Comparison-Validity of Manuscript Title

For the reasons described above we think that the serial passaging gives us a good starting point to capture regulators of stem cell function. We regarded this first experiment as a screening exercise where we sought to capture as many potential regulators of stem cell function as possible and then validate these in early passage keratinocytes which had not been passaged many times.

Our work aimed to identify transcription factors which regulated stem cell function, in the case of IRF2 which resulted in the loss of stem cell function. In epidermis, as with several other epithelial tissues with high renewal rates, there is an increasing appreciation that cells are much more plastic than previously described and that “hard-wired” stem cells are not clearly recognizable. This concept is covered in detail in a recent review by Hans Clevers and Fiona Watt (Defining Adult Stem Cells by Function, not by Phenotype, Annual Review Biochemistry 87, p1015-1027, 2018). Such concepts support the idea that stem cell renewal is best defined on a population level and that the ability of a tissue to self-renew or repair itself over the lifetime of an organism is what is physiologically relevant. Such concepts of stem cell renewal make it appropriate to investigate stem cell function in a population of keratinocytes for physiological relevance to tissue function in homeostasis and repair. Therefore we believe that the title of our manuscript remains valid when editing a population of keratinocytes and obtaining cells with greater stem cell function.

We accept the concerns of reviewer 1 that we have not sufficiently characterized HSCP and LSCP populations and have now provided significant additional data and analysis as described under the specific questions below.

Additional points:

- In Fig 1 the authors should better characterize the HSCP and LSCP they are using. For example, they should evaluate holoclones/meroclones and paraclones between the two populations. Since the holoclone are generated only by somatic stem cells/precursors, it will be crucial to see that the LSCP are indeed able to produce holoclones. This is also important in the experiment shown in Fig 5C (also 5A and B).

This is a good point, as the most critical experiment is whether IRF2 removal actually enhances the stem cell characteristics of the population. As requested, we have now provided data on the number of holoclones, meroclones and paraclones formed with clonogenic seeding of control keratinocytes and IRF2 edited keratinocytes (Figure 5E) with some representative images to guide the reader on the morphological differences between these types of colonies (Figure 5E). We find that IRF2 edited cells indeed show a greater ability to form holoclones in line with our contention that they are more stem-cell like (see new Figure 5E). The text on Page 12/13 (again highlighted in yellow) has been amended to incorporate and explain this new data. We also provide some further clarification of the findings from the 3D skin model.

- Cellular senescence, beside being associated to an increase of beta-galactosidase staining, is also associated to other markers: expression of p16, PML, hypo-phosphorylation of Rb, decrease of telomerase activity, accumulation of DNA damage. How the LSCP cells express this markers in comparison with HSCP? Does this markers are altered in relation to IRF2 expression? Specifically, LSCP keratinocytes, after IRF2KO, have reduced senescence markers? Do they form holoclones?

We now include P16 Western blot as requested (new Figure 7A). Firstly this shows that the p16 expression is higher in LSCP-HKs than HSCP-HKs as expected. Secondly and specifically requested by the reviewer, we show that IRF2 KD in LSCP-HKs shows a reduction in the senescence marker p16 (new Figure 7A). Interestingly IRF2-KD can further reduce the low expression in the control HSCP-HKs (new Figure 7A). We have not analyzed holoclones in this particular

experiment but have shown increased holoclone formation by IRF2 KD in the previous experiments described in Figure 5. The manuscript text has been edited to incorporate these changes –see page 12/13 highlighted yellow text.

- The reduced migration capacity observed in the LSCP cells is a direct effect of IRF2 modulation? Could be p63 down-regulation been responsible for this? Is it reported by many articles that p63 controls a set of genes related to cell adhesion and cell-cell contact.

Our experiments demonstrate that IRF2-KD cells show enhanced migration compared to control cells but do not identify the molecular mechanism. Our contention is that IRF2 is regulating a transcriptional circuitry which is driving a loss of stem cell function in keratinocytes. This transcriptional circuitry involves a complex network of multiple transcription factors which in keratinocytes will include p63. In figure 1, p63 is identified from a Metacore analysis in the high stem cell potential keratinocytes but not in the low stem cell potential which suggests that the role of p63 is downregulated in low stem cell potential cells which in turn may have a role in the reduced cell adhesion. P63 is certainly reported by many articles to regulate cell-cell and cell-matrix in squamous epithelial cells such as keratinocytes. A direct mechanistic link between IRF2 and p63 cannot be inferred from our work and would need further study.

- The majority of the 3D skin images shown are low quality (ie. 5B, out of focus and/or damaged). In general, measurement of the 3D skin thickness is not sufficient to claim an increase in stem cell potential. The authors should also evaluate the “quality” of the 3D reconstructed skin. For instance, they should evaluate basal layer/upper layers markers, Ki67+ cells, p63+ cells, evaluate if the switch between proliferating/differentiating layer (K14/K10) occurs normally in the different conditions shown (both for Fig 5 and Fig 7), etc.

Image Quality: We accept that the quality of the 3D skin images in Figure 5 could be improved and we now provide better images for the SNAI2 and YY1 edited cells (Figure 5B) and for IRF2-KD and control cells (Figure 5D). We have added a label # to show the thickness of the epidermis in all figures.

Epidermal thickness:

Previous studies have shown a good link between stemness, reduced replicative senescence, enhanced proliferation and formation of a thicker epidermis with better differentiation (Jobeili et al, Selenium preserves keratinocyte stemness and delays senescence by maintaining epidermal adhesion). This reference and explanation is now added to the text (P12). However, we agree with the reviewer that epidermal thickness in the 3D model is insufficient to claim increased stem cell potential and therefore we also provide data on clonogenicity (including the requested holoclone analysis) and migration to support our claim that stem cell potential is raised in IRF2KO cells. As requested, we now provide further analysis of the quality of the epidermis and higher quality images of all 3D constructs in Figure 7. In the original manuscript we described a greater cellularity of constructs assembled from IRF2-KD cells, we now provide quantification of that difference. Therefore in Figure 7E alongside the histogram on epidermal thickness, we include a second histogram detailing the number of nuclei/mm of construct (Figure 7E).

Ki67+ cell staining was not performed in this experiment as previous experience with our 3D model shows that at Day 1 and 3 we have significant numbers of Ki67 positive cells but minimal positive cells by Day 11. As our constructs were sampled between Day11 and Day 12, very few Ki67+ cells would have been present so comparisons of conditions would not have been possible. We include some of our data below to demonstrate this point but do not include this in the manuscript as we have not analyzed Ki67 staining in our experiments.

- Fig7 F is not clear at all. Based on my experience this experiments need to be re-done. Why they included the dotted line? The authors should add K14 staining as basal layer marker. Why a dark staining appear in the derma? Is the antibody specific? Why LSCP do not form the cornified layer (also in Fig 7E)? Again, without specific markers for each layer is not possible to evaluate what is shown in the 3D skin culture in all the experiments presented.
- It has been demonstrated that IRF2KO give a psoriatic-like phenotype in mouse skin. Is the phenotype observed in IRF2KO keratinocytes could be due to production of cytokines?

Higher quality images have been provided for Figure 7 and further explanation provided in Figure 7 figure legend which clarifies the data significantly. Our use of the dotted line was inadequately explained and this has also been resolved by using 2 dotted lines to identify the basal line with a full description in new Figure 7 legend. The stratum corneum can be seen in H&E stained sections as a pink layer identified as layer 5 in the new Supplementary Figure 16. This new Supplementary Figure 16 gives much greater detail of the different layers within our 3D constructs with quantification of the cornified layer (stratum corneum) and also the stratum spinosum layer. We believe that this should resolve the questions around the differential histology of the 3D human skin constructs.

For the question around the possible connection of our LSCP-HKs to the psoriasis phenotype. Our data showed an upregulation of many psoriasis related genes in LSCP-HKs which were reversed towards normal with IRF2-KD. Our data is in contradiction to what has been observed in IRF2-KO mice skin, where IRF2 KO induced psoriatic type inflammation. We have no data around the involvement of specific cytokine release in this study, but their involvement would be plausible as senescence is accompanied by expression of a pro-inflammatory phenotype. The only similarity to the mice study is that IRF2-KO mice skin showed hyperplasia suggesting a similar role for IRF2-KD in human keratinocytes.

Reviewer #2 (Remarks to the Author):

This is a well performed study to identify core regulatory circuitry (CRC) that dictates cell identity in primary keratinocytes with high stem potential (HSCP-HK) and those with low stem cell potential (LSCP-HK). The unbiased approach used to identify CRC TFs is novel and rigorous, and the findings novel with potentially high impact in the field of epidermal biology. Comments/questions that I have for the authors are mostly minor and are listed below:

1. The model used to establish LSCP-HKs, using serial passaging, appears to recreate the phenotype/function of LSCP-HKs, but the distinction between LSCP-HKs and simply senescent cells, if in fact there is a distinction, should be better explained in the text. Namely, in normal human epidermis, is the process of differentiation, stratification, and eventual death of keratinocytes identical to or at least highly overlapping with the process of cellular senescence? How confident are we that the serial passaging model is truly mimicking the in vivo process of keratinocyte differentiation as these cells rise above the basal layer? Note to authors that these concerns are largely assuaged by the transcriptome/epigenome/CRISPR data presented, and by the reversibility of LSCP-HK function with IRF2 KO, so simply a better explanation of how differentiation relates to senescence would be appreciated to those who are not epidermal biologists.

These are good points which have been extensively addressed in response to Reviewer 1 comments. We have made significant amendments to the manuscript to explain our use of replicative senescence as a model system. Senescence widely correlates with differentiation of keratinocytes and therefore we consider this a valid model as a starting point for this study. The results from the transcriptional profiling data support this contention as we show that with acquisition of senescent markers there is a concomitant increase in differentiation markers in the LSCP-HKs.

2. Fig. 3A: What cells are being analyzed? This information does not appear to be stated anywhere in the text, methods, figure, or figure legend. Was the bioinformatic analysis performed on HSCP-HKs only?

We apologize for this oversight. In the analysis outlined in Figure 3A, we analyze the union of active genes and H3K27ac sites in both HSCP-HKs and LSCP-HKs in order to create the overall active cis-regulatory landscape for HKs. In subsequent analysis, we then examine which trans factors and cis-regulatory elements within the HK regulatory landscape show LSCP or HSCP specific activity. We have amended Fig. 3A to reflect this more clearly.

3. Page 11, line 9: How do the authors justify focusing most of their studies on IRF2 as the master regulator? For example, BRD2 appears also to be a master regulator based on the data. Is this because BRD2 is not a TF, or because it does not alter migratory properties?

We did perform some preliminary follow-up work with BRD2 and although we did see increased keratinocyte clonogenicity with BRD2-KD, we did not see any change in cell migration. We were not sure whether further changes would be observed with time as the CRISPR mini-pool screen was performed over 45 days. Clonogenic assays are long term to allow colony formation and could explain why we captured differences with this assay and not the short term migration assay. In parallel, we were observing robust and reproducible phenotypic changes with IRF2-KD and as considerable experimentation was required for the final validation of each gene, we decided to focus our efforts on IRF2. This point and the reasons for our decision to focus on IRF2 are now clarified in the manuscript-see edited highlighted text on page 11. We have also added a reference to a recently published paper by Slivka et al (2019) linking BRD2 to IL-17 signaling and psoriasis.

4. Fig. 5D, 7E and 7F: Some increased descriptive input from your pathologist might help the text on p12 (second paragraph). The authors describe differences in the epidermal component based on rather basic features such as "thickness", "cellularity", and "organization". What differences are present in the various layers? Have the authors scored the thickness of the stratum corneum in the various conditions? One might expect with greater differentiation that the corneal layer would be thinner perhaps in the IRF2 knockouts, though it does not appear to be. Some discussion on this point might help. Additionally, the stratum(s) between the basal and corneal layers appear less differentiated (more "cellular" on the words of the authors) in the IRF2 knockouts than CT counterparts. One might express this in terms of nuclear/cytoplasmic ratio, or nuclei/unit area, and likely this could be quantified. An attempt at illustrating differentiation with Keratin 10 is made in Fig. 7F, however it is not easy to see with the H&E counterstain. This would be better visualized with a minimal (e.g. hematoxylin-only) counterstain.

These are again very good points which had also been raised by Reviewer 1. We accept that the image quality and analysis of the 3D model presented in the original manuscript could be improved. As described above we have provided higher quality images, additional analysis and explanation of our findings which we hope will satisfy the concerns of the reviewer.

5. Page 14, paragraph 2: It is hard to interpret what the authors are saying. They state that IRF2KO in mice has a psoriatic phenotype, however it appears that they are saying that IRF2 KO in this model system reverts LSCP-HK cells to a less psoriatic phenotype. This seems contradictory and some clarification here is needed.

IRF2-KO is described to generate epidermal hyperplasia which correlates with the keratinocyte phenotype that we see in vitro. This hyperplasia generates a "psoriasis -like" inflammation but this may describe the infiltration of immune cells and not reflect the epidermal pathology. Therefore we think that the contradiction might result from comparison of profiling data on pure keratinocyte populations in vitro with the much more complex situation in mouse epidermis where hyperplasia is leading to an imbalance of the stem/differentiated compartments potentially leading to inflammation.

6. Page 18, line 6: It is interesting that IRF2 transcript levels do not differ between LSCP-HK and HSCP-HK cells given data that suggests an important anti-stemness role. If a good antibody to IRF2 exists, IHC or IF would be able to show whether IRF2 protein levels change as keratinocytes differentiate from basal to upper layers in their epidermal-dermal 3D model.

We have shown by western blot that IRF2 protein expression does not significantly change in keratinocytes from early to late passage. Therefore we did not consider that further IHC demonstration would be useful. Our data support a re-wiring of the transcriptome in LSCP-HKs where differential chromatin localization of IRF-2 rather than amount of IRF-2 is driving the phenotypic change.

Reviewer #3 (Remarks to the Author):

The manuscript "IRF2 is a master regulator of human keratinocyte stem cell fate" by Mercado et al. identifies increased IRF2 expression as an antagonist of stem cell function in human keratinocytes. The manuscript is well written with appropriate experiments performed and presented. A few minor concerns that could be addressed to strengthen the manuscript are listed below:

1. Since a comparison between human and mice IRF2 is made, it would be informative to state what the % conservation of IRF2 is, and what the conservation means (or does not mean)

There is significant conservation between mouse and human IRF-2 (about 93% similar according to string). The biggest difference in amino acid sequence can be found in the “Repressor Domain” (see below) which could explain some differences between the roles of IRF2 in mouse and human. We can add this information to the manuscript upon reviewer 3 recommendation.

Irf2-M	1	MPVERMRMRPWLEEQINSNTI PGLKWLNKEKKIFQIPWMHAARHGWDVE MPVERMRMRPWLEEQINSNTI PGLKWLNKEKKIFQIPWMHAARHGWDVE
IRF2-H	1	MPVERMRMRPWLEEQINSNTI PGLKWLNKEKKIFQIPWMHAARHGWDVE
Irf2-M	50	KDAPLFRNWA IHTGKHQPGIDKDPDKTWNKAFRCAMNSLPDIEEVKDRS KDAPLFRNWA IHTGKHQPG+DKPDKTWNKAFRCAMNSLPDIEEVKD+S
IRF2-H	50	KDAPLFRNWA IHTGKHQPGVDKDPDKTWNKAFRCAMNSLPDIEEVKDKS
Irf2-M	99	IKKGNNAFRVYRMLPLSERPSKKGKPKTEKEERVKHIKQEPVESSLGL IKKGNNAFRVYRMLPLSERPSKKGKPKTEKE++VKHIKQEPVESSLGL
IRF2-H	99	IKKGNNAFRVYRMLPLSERPSKKGKPKTEKEDKVHIKQEPVESSLGL
Irf2-M	148	SNGVSGFSPEYAVLTS AIKNEVDSTVNI IIVVGQSHLDSNIEDQEIVTNP SNGVS SPEYAVLTS IKNEVDSTVNI IIVVGQSHLDSNIE+QEIVTNP
IRF2-H	148	SNGVSDLSPEYAVLTSTIKNEVDSTVNI IIVVGQSHLDSNIENQEIVTNP
Irf2-M	197	PDICQVVEVTTESD DQPVSMSELYPLQISPVSSYAESSETTDSVASDEEN PDICQVVEVTTESD+QPVSMSELYPLQISPVSSYAESSETTDSV SDEE+
IRF2-H	197	PDICQVVEVTTESD EQPVSMSELYPLQISPVSSYAESSETTDSVPSDEES
Irf2-M	246	AEGRPHWRKRS IEGKQYLSNMG TRNTYLLPSMATFVTSNKPDQLQVTIKE AEGRPHWRKR+IEGKQYLSNMGTR +YLLP MA+FVTSNKPDQLQVTIKE
IRF2-H	246	AEGRPHWRKRNI EGGKQYLSNMG TRGSYLLPGMASFVTSNKPDQLQVTIKE
Irf2-M	295	DSCPMPYNSSWPPFDLPLPAPVTPTPSSSRPDRETRASVIKKTSDITQ +S P+PYNSSWPPF DLPL + +TP SSSRPDRETRASVIKKTSDITQ
IRF2-H	295	ESNPVPYNSSWPPFQDLPLSSSMTPASSSRPDRETRASVIKKTSDITQ
Irf2-M	344	ARVKSC
IRF2-H	344	ARVKSC

Irf2-M: Mouse IRF2

IRF2-H: Human IRF2

DBD: DNA-Binding Domain

IAD: IRF-associated Domain

RP: Repressor Domain

Different amino acid between human and mouse IRF2

2. “Some of the top predicted TFs did not confirm in the screen most likely due to technical limitations with CRISPR editing of primary cells, current chromatin profiling techniques and assumptions required to build the core transcriptional

algorithm.”—This statement is unclear and confusing and requires some elaborate explanation regarding the indicated limitations.

We have clarified this section in the text explaining more clearly why some of the assumptions needed to generate the core transcriptional circuitry and the technical limitations of CRISPR-Cas9 editing may have led to a lack of confirmation for some of the top predicted TFs. Edited text is shown highlighted in yellow on page 11.

3. It would be interesting to see the effects of depleting both YY1 and SNAI2 simultaneously in the experimental systems described.

YY1 and SNAI2 depleted keratinocytes start to resemble late passage cells very quickly after CRISPR-Cas9 editing, in the case of YY1 within a couple of days. Therefore, this proposed experiment is unlikely to be possible as the resultant keratinocytes wouldn't remain viable for long enough to analyze their phenotype.

4. It will be appropriate to provide the wound healing assay images along with quantification in Fig 1C.

As requested, images for the HSCP-HKs and LSCP-HKs at the 8 hour time point have been added to Figure 1C.

5. In Fig 1C Y-axis - 'relative wound density' term is confusing. Should it be wound healing capacity or % migration?

For the migration assay, we make a “wound” in an intact monolayer with a commercial tool at a defined location in the well which is then recognized by the Image Software package supplied with the Incucyte-Zoom (Essen Bioscience). This software measures the density of keratinocytes within the wound in comparison to the keratinocytes in the undamaged monolayers either side of the wound. This is called the relative wound density. We have added some additional wording to the text to clarify this read-out. Below is the manufacturer's description of the meaning of relative wound density. Again we are happy to include this in the manuscript if this is thought useful.

Relative Wound Density (%): This metric relies on measuring the spatial cell density in the wound area relative to the spatial cell density outside of the wound area at every time point. It is designed to be 0% at t=0, and 100% when the cell density inside the wound is the same as the cell density outside the initial wound. In this respect, the metric is self-normalizing for changes in cell density which may occur outside the wound due to cell proliferation and/or pharmacological effects. Importantly, the RWD metric is robust across multiple cell types as it does not rely on finding cell boundaries.

REVIEWERS' COMMENTS:

Reviewer #1 (Remarks to the Author):

The authors addressed the majority of the referee concern.

Reviewer #2 (Remarks to the Author):

The authors have adequately addressed concerns of all reviewers, including myself. Methods provided include sufficient detail to reproduce. Statistical analysis is appropriate.

Overall, a rigorous, data-rich, unbiased approach was used and adequately validated to produce the following findings: the identification of the core transcriptional regulatory circuitry that defines keratinocyte stemness, with the identification of pro-stem potential factors, SNAI2 and YY1, and anti-stem potential factor, IRF2. The findings are novel with high impact potential in epidermal biology. One comment below:

IRF2 and SNAI2/YY1 appear to be necessary for anti-stemness and pro-stemness, respectively, in keratinocytes. I don't think the current study demonstrates that they are sufficient, as there are no gain-of-function studies (i.e. forced expression of IRF2 to induce low stemness in otherwise high stem potential cells). The authors might consider adding this point to their discussion.

Response to Reviewer's Comments:

REVIEWERS' COMMENTS:

Reviewer #1 (Remarks to the Author):

The authors addressed the majority of the referee concern.

No further edits required

Reviewer #2 (Remarks to the Author):

The authors have adequately addressed concerns of all reviewers, including myself. Methods provided include sufficient detail to reproduce. Statistical analysis is appropriate.

Overall, a rigorous, data-rich, unbiased approach was used and adequately validated to produce the following findings: the identification of the core transcriptional regulatory circuitry that defines keratinocyte stemness, with the identification of pro-stem potential factors, SNAI2 and YY1, and anti-stem potential factor, IRF2. The findings are novel with high impact potential in epidermal biology. One comment below:

IRF2 and SNAI2/YY1 appear to be necessary for anti-stemness and pro-stemness, respectively, in keratinocytes. I don't think the current study demonstrates that they are sufficient, as there are no gain-of-function studies (i.e. forced expression of IRF2 to induce low stemness in otherwise high stem potential cells). The authors might consider adding this point to their discussion.

A sentence addressing this point in the context of IRF2 has been added to the discussion as requested. "These findings support a major role for IRF2 in driving loss of keratinocyte stem cell function although this was not directly demonstrated with overexpression experiments"